# Globally scalable glacier mapping by deep learning matches expert delineation accuracy

Konstantin A. Maslov [1] ✉, Claudio Persello [1], Thomas Schellenberger [2] & Alfred Stein [1]

Accurate global glacier mapping is critical for understanding climate change impacts. Despite its importance, automated glacier mapping at a global scale remains largely unexplored. Here we address this gap and propose Glacier-VisionTransformer-U-Net (GlaViTU), a convolutional-transformer deep learning model, and five strategies for multitemporal global-scale glacier mapping using open satellite imagery. Assessing the spatial, temporal and cross-sensor generalisation shows that our best strategy achieves intersection over union >0.85 on previously unobserved images in most cases, which drops to >0.75 for debris-rich areas such as High-Mountain Asia and increases to >0.90 for regions dominated by clean ice. A comparative validation against human expert uncertainties in terms of area and distance deviations underscores GlaViTU performance, approaching or matching expert-level delineation. Adding synthetic aperture radar data, namely, backscatter and interferometric coherence, increases the accuracy in all regions where available. The calibrated confidence for glacier extents is reported making the predictions more reliable and interpretable. We also release a benchmark dataset that covers 9% of glaciers worldwide. Our results support efforts towards automated multitemporal and global glacier mapping.

Glaciers are highly sensitive to alterations in temperature and precipitation, making them important indicators of climate change, and recognised as an essential climate variable within the Global Climate Observing System (GCOS) programme[1]. In the last decades, the vast majority of glaciers worldwide have diminished in size and are expected to continue retreating, and 49 ± 9% of glaciers are likely to disappear by 2100 regardless of the greenhouse gas emission scenario[2–4]. This loss in glacier mass contributes significantly to rising sea levels, accounting for approximately 25–30% of the observed increase since 1961, with regions such as Alaska, the Canadian Arctic, the periphery of Greenland, the Southern Andes, the Russian Arctic and Svalbard having the greatest shares[5]. The projected rise in sea level from glaciers (90 ± 26 to 154 ± 44 millimetres sea level equivalent) poses a severe threat to millions of households predicted to be below

high-tide lines by the end of the century[4,6]. In addition, glacier run-off can compensate for seasons of low flow and offset water shortages during droughts[7,8]. Glacier retreat affects hydroelectric power generation[9], drinking water quality, agricultural productivity[10], ecosystems and biodiversity[11] and is related to glacial lake outburst floods with possibly devastating consequences[12], though their severity tends to diminish over time[13].

Regularly updated glacier inventories, which track glacier area as one of the GCOS essential climate variable products[1], provide valuable information to measure, predict, mitigate or adapt to these challenges. Yet currently, the GSOC standards for estimating regional glacier area change in most glacierised regions on a decadal scale are not fulfilled[1]. Furthermore, existing glacier inventories currently have several limitations. Regional and sub-regional inventories have restricted spatial

[1]Department of Earth Observation Science, Faculty of Geo-information Science and Earth Observation (ITC), University of Twente, Overijssel, The Netherlands. [2]Department of Geosciences, Faculty of Mathematics and Natural Sciences, University of Oslo, Østlandet, Norway. ✉e-mail: k.a.maslov@utwente.nl

coverage and lack consistency in the methods and principles employed for deriving glacier outlines. The global Randolph Glacier Inventory (RGI) has a limited temporal coverage with most glacier outlines mapped around the year 2000. It serves as a baseline dataset in many glaciological studies[4,6,14], but its use is limited when trying to understand temporal evolution and more recent changes[15]. Similarly to the regional inventories, the Global Land Ice Measurements from Space (GLIMS) database, the largest inventory database and a superset of RGI, despite being global and multitemporal, suffers from data quality issues as it comprises a compilation of regional inventories with varying levels of accuracy and consistency. This makes it challenging to obtain a comprehensive and reliable view of glaciers worldwide at a multitemporal level[16]. These limitations have significant scientific effects. Glaciologists who derive critical glacier products, such as ice surface velocities, ice thickness and geodetic mass balance, depend on accurate glacier outlines[6,14]. Any inaccuracies in these outlines may propagate errors to downstream tasks affecting subsequent analyses and applications, e.g., global mass balance modelling studies[4]. Moreover, glacier outlines serve as essential inputs for modelling efforts, allowing scientists to make informed assumptions about physical processes and to forecast the evolution of glaciers. The availability of consistent, multitemporal glacier outlines, given they match or exceed the quality of existing inventories, would not only improve the accuracy of future glacier datasets and studies but would also offer a more reliable basis for calibrating glacier models to past periods. They will also lead to enhancements in glacier evolution model development, e.g., allowing for the incorporation of regularly updated glacier geometries into glacier dynamics models as well as the incorporation of calving dynamics of marine-terminating glaciers and a better representation of extreme changes in areas, e.g., via advances through glacier surges. Overall, we expect multiple methodological innovations which will enhance our ability to better constrain past and predict future glacier evolution. Producing consistent inventories, however, is a highly challenging and time-consuming task often requiring extensive visual interpretation and manual digitisation of satellite images by experts. Thus, a crucial long-term goal is to automate glacier mapping globally and across multiple time periods in a consistent manner.

Simple thresholds of optical band ratios are effective methods to map clean-ice glaciers on different scales[17–19]. The choice of threshold values, however, can vary for different imaging conditions[20] and within a scene[17]. Moreover, threshold-based methods ignore spatial patterns and textural information and, thus, fail to classify complex glacier parts such as debris- or vegetation-covered ice requiring labour-consuming manual corrections or application of more sophisticated methods[21]. To overcome some of these limitations, researchers have focused on incorporating additional data to optical imagery such as thermal-infrared bands[22], interferometric synthetic aperture radar (SAR) data[18,23,24] as well as SAR amplitude time series[25]. Alifu et al.[26] applied machine learning to multi-source data combining optical, SAR, thermal data and a digital elevation model (DEM) and found random forests to have good correspondence with manually derived outlines.

In recent years, deep learning has been applied to glacier mapping. GlacierNet, a modification of SegNet[27], was utilised in the Karakoram and Nepal Himalayas leveraging both optical and topographical features[28]. Subsequently, Xie et al.[29] compared GlacierNet with other convolutional models, where DeepLabv3+[30] demonstrated superior performance. Yan et al.[31] proposed a spatiospectral attention module and combined it with U-Net[32] to map glaciers on a sub-regional scale in the Nyainqentanglha Range, China. Recently, following the introduction of vision transformers (ViTs)[33], which are capable of extracting long-range dependencies in images due to the global attention mechanism, transformers were adapted for different computer vision tasks including semantic image segmentation[34–36]. They were also applied in remote sensing often introducing hybrid convolutional-transformer models that aim at leveraging the advantages of both types of architectures[37,38]. One such hybrid model for ice mapping was tested in the Qilian mountains, China[39]. Despite the variety of methods, none of them have been applied and validated in mapping glaciers on a large or global scale, thus limiting their transferability in time and space.

Mapping glaciers on a global scale presents unique challenges, particularly in achieving model generalisation across diverse environmental conditions, time ranges and satellite sensors. While it may be relatively straightforward to map glaciers in a single region and year, extending this to a worldwide scale adds multiple dimensions of complexity. First, the sheer volume of data necessitates efficient data engineering and management. Second, reaching global generalisation highlights critical aspects of data preparation and quality as well as their representativeness. Furthermore, no attempt has been made to quantify the automated mapping uncertainty of the glacier outlines despite its crucial role in enhancing map reliability, facilitating time series analysis and enabling transparent communication of results generated with artificial intelligence (AI) to stakeholders and decision-makers. Additionally, as we will show, uncertainty quantification can be used to estimate mapping quality in the absence of reference data. We address these gaps in this paper.

This study makes several contributions to the field of global glacier mapping. We introduce a convolutional-transformer deep learning model and release a comprehensive multimodal dataset (approximately 400 GB) leveraging the potential of using open optical and SAR satellite data in a single semantic segmentation framework. The dataset spans diverse glacierised environments with 9% of all glaciers worldwide included. We explore five deep learning–based strategies that aim to achieve high generalisation across regions, satellite sensors and through time. Our results suggest that one can expect an intersection over union ($IoU = |T \cap P|/|T \cup P|$, where $T$ and $P$ are the reference and predicted areas, respectively) score of 0.85 or higher on previously unobserved satellite imagery on average. This, however, might drop to >0.75 for debris-rich and increase to >0.9 for clean-ice–dominated areas. Comparing the proposed deep learning approach to human expert uncertainty in terms of area and distance shows it approaches or matches expert delineation in most cases while ensuring objective, scalable and reproducible results, which are necessary for quantifying glacier area changes over decades. Additionally, our predicted confidence analysis reveals valuable insights into the model behaviour and limitations, enhancing the reliability of the predictions as well as providing a means to correct the outlines for glaciers of specific interest. While challenges persist, such as the identification of debris-covered tongues, ice mélange, snow patches and glaciers in mountain shadows, our findings represent a significant step forward towards automated global glacier mapping, offering improved accuracy and more efficient glacier inventory generation.

## Results

We compiled two comprehensive datasets from publicly accessible satellite images and glacier inventories—a tile-based dataset and an independent acquisition test dataset. The tile-based dataset (Fig. 1), primarily used for the development and preliminary testing of the model, covers 11 diverse regions globally and spans from 1988 to 2020. It is structured into non-overlapping near-squared 10 × 10 km² tiles. The tiles were randomly split into training, validation and testing subsets. This dataset contains a broad spectrum of glacier types across different environmental settings and includes optical, SAR and DEM features and reference data from GLIMS[16], RGI[15] and regional inventories[40–43] for both training and accuracy estimation. The data were organised into three tracks—(1) Optical+DEM, (2) Optical+DEM +thermal and (3) Optical+DEM+InSAR—to investigate the influence of different feature sets on the classification performance. The independent acquisition test dataset is specifically used to test the generalisation of our model over different satellite image acquisitions in a

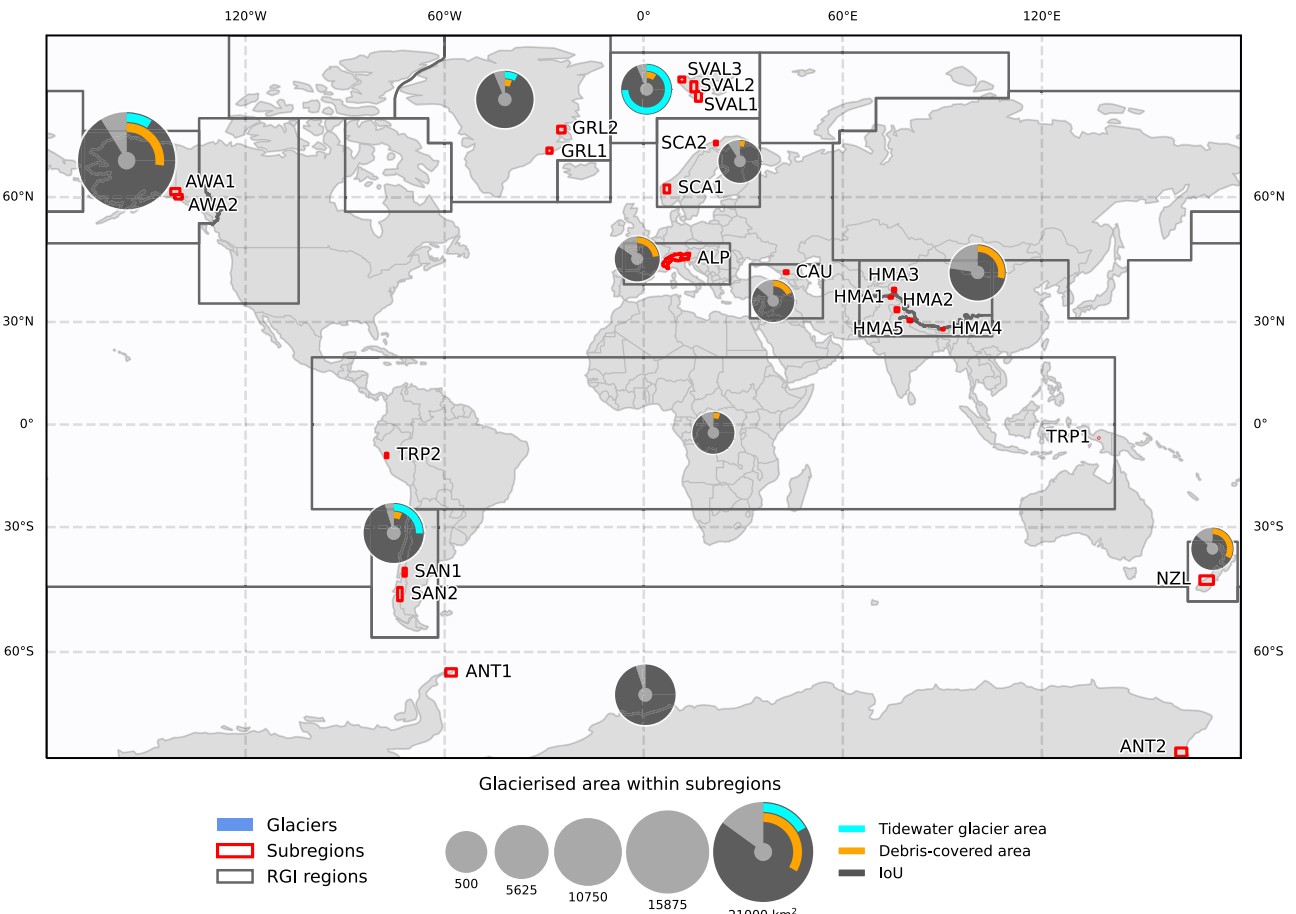

**Fig. 1 | Tile-based dataset and results overview.** Region abbreviations: ALP (the Alps), ANT (Antarctica), AWA (Alaska and Western America), CAU (Caucasus), GRL (Greenland), HMA (High-Mountain Asia), TRP (low latitudes), NZL (New Zealand), SAN (the Southern Andes), SCA (Scandinavia), SVAL (Svalbard). The glacier outlines and tidewater glacier areas are based on RGI7.0[15]. Debris coverage is adapted from Herreid and Pellicciotti[79]. The intersection over union (IoU) values are presented for the GlaViTU model trained globally with Optical+DEM data. The statistics for Central, South West and South East Asia are aggregated. Source data are provided as a Source Data file.

"real-world" application scenario and includes data from the Swiss Alps, Scandinavia, Alaska and Southern Canada. This test dataset enabled more challenging temporal and spatial generalisation assessments of the models trained solely on the tile-based dataset. For more details on the data, please see the "Study areas and datasets" subsection.

We propose Glacier-VisionTransformer-U-Net (GlaViTU, Supplementary Fig. 10), a deep learning model, which is designed to capture both global and local image patterns (see the details in the "GlaViTU" subsection) and trained it on the tile-based dataset. Five strategies to achieve high generalisation were explored: (1) a global strategy that implies one model trained over all regions, (2) a regional strategy that trains one model per one region or a group of regions (e.g., sharing similar characteristics such as debris cover percentage), (3) a finetuning strategy in which the global model is finetuned for every region (or a group of regions) individually and (4) region encoding and (5) coordinate encoding strategies where we feed location data, respectively, one-hot encoded regions and geographical coordinates, directly into the model. Coupled with the region encoding strategy, we also implemented bias optimisation that aimed at reducing the domain shift by adjusting biases introduced by the location data during inference (see the details in the "Strategies towards global glacier mapping" subsection). We tested these strategies on both the tile-based and independent acquisition test datasets. In addition, we derived predictive confidence from Monte-Carlo dropout[44] and plain softmax scores[45], performed calibration to better align confidence with actual accuracy for more reliable interpretation and compared those, as well as explored which targets exhibit low confidence scores (see the details in the "Uncertainty quantification" subsection). For a detailed description of what metrics we used to evaluate our results, please see the "Accuracy assessment" subsection.

## Model performance

GlaViTU, trained globally on Optical+DEM data, achieved a high accuracy with an average IoU of 0.894 (Supplementary Table 2). The model performed very well for regions with predominantly snow and ice conditions such as the Southern Andes (IoU = 0.952), Antarctica (0.949), Greenland (0.937) and Svalbard (0.936). A drop in performance is seen for areas with significant debris cover, with the worst performance in High-Mountain Asia (0.774), but also for Caucasus (0.862) and the Alps (0.844). Several classified tiles are shown in Supplementary Fig. 2.

GlaViTU still faced challenges in certain scenarios (Supplementary Fig. 5). For instance, identifying some debris-covered tongues could be challenging in some cases, the model tended to overpredict debris (Supplementary Fig. 5a) and to miss it in other cases (Supplementary Fig. 5b). Also, GlaViTU struggled with shadowed ice, sometimes failing to classify it accurately (Supplementary Fig. 5c). In situations of dense ice mélange, GlaViTU yielded a significant amount of false positives (Supplementary Fig. 5d, e). Also, there could be unexpected artefacts on coastlines (Supplementary Fig. 5f).

**Table 1 | Independent acquisition test results**

| Region | IoU[a] of different strategies (GlaViTU) | | | | | IoU of band ratio |
|---|---|---|---|---|---|---|
| | Global | Regional | Finetuning | Region encoding | Coordinate encoding | |
| Temporal/cross-sensor generalisation | | | | | | |
| Swiss Alps | 0.865 | 0.851(ALP) | 0.861(ALP) | 0.856(ALP) | 0.545 | 0.772 |
| | | 0.857(AWA+ALP+CAU) | 0.862(AWA+ALP+CAU) | **0.868**[b] | | |
| Southern Norway (SCA1) | 0.926 | 0.921(SCA) | 0.918(SCA) | 0.928(SCA) | 0.876 | 0.915 |
| | | | | **0.933**[b] | | |
| Spatial generalisation | | | | | | |
| Alaska | **0.914** | 0.809(AWA) | 0.894(AWA) | 0.902(AWA) | 0.253 | 0.852 |
| | | 0.890(AWA+ALP+CAU) | 0.904(AWA+ALP+CAU) | 0.912[b] | | |
| Southern Canada | 0.870 | 0.808(AWA) | **0.870**(AWA) | 0.867(AWA) | 0.858 | 0.843 |
| | | 0.848(AWA+ALP+CAU) | 0.864(AWA+ALP+CAU) | 0.870[b] | | |

Parentheses specify the regions used for regional/finetuned models and the flags for region encoding: ALP (the Alps), AWA (Alaska and Western America), CAU (Caucasus) and SCA (Scandinavia).
[a]The best IoU values are in bold.
[b]The use of bias optimisation.

To sum up, GlaViTU showed high performance in various settings of glacier mapping. Challenges still remain such as debris-covered tongue identification, shadowed ice detection, errors in dense ice mélange regions and artefacts at coastlines.

## Comparison of the strategies towards global mapping on the tile-based dataset

We tested five strategies towards global glacier mapping and evaluated their performance. Supplementary Table 4 provides an overview of the results. Overall, the regional and finetuning strategies emerged as the most promising with average IoUs of 0.902 and 0.901, respectively, delivering the best models for all but two regions. The region encoding strategy performed slightly worse on average (IoU = 0.897). It showed a slight performance gain as compared to the global strategy (IoU = 0.894) and performed better in all but one region. The coordinate encoding strategy had the lowest average IoU of 0.893 and did not perform best in any of the regions. Although, on average, the differences are minor, and it outperformed some of the other strategies in some regions, it generally does not offer performance gains for global glacier mapping.

A qualitative analysis of the performance of the five strategies in different scenarios allowed the identification of several patterns. Some classification results derived with different strategies are shown in Supplementary Fig. 11. Across strategies, the results for clean ice classification were almost identical (Supplementary Fig. 11a, b), while the regional and finetuning strategies were superior in locating glacier tongues in more challenging settings (Supplementary Fig. 11c, d). Both regional and finetuning strategies tended to produce fewer false positives for ice mélange (Supplementary Fig. 11e, f) and solved the issues with the coastline artefacts (Supplementary Fig. 11g), further enhancing the reliability and precision of the mapping outcomes.

In summary, when tested on the tile-based dataset, the regional and finetuning strategies provided the best mapping accuracy with notable improvements in challenging settings.

## Independent acquisition tests

When assessing the strategies in terms of their temporal and spatial generalisation performance using the independent acquisition test data, the results diverged from the initial findings, as shown in Table 1. For instance, the region encoding strategy coupled with bias optimisation yielded the best mapping quality on average (IoU = 0.896), while the achieved performance gain remained relatively modest when compared to the global strategy (IoU = 0.894). The regional and finetuning strategies did not maintain their superiority in these tests. The only exception was Southern Canada, where the finetuning strategy

marginally outperformed the other strategies, but we consider this difference of minor importance. On average, the finetuned models exhibited noticeable improvements as compared to the regional models. Additionally, training and finetuning models for clusters of regions with similar characteristics yielded an overall enhancement in performance as compared to a single region. The coordinate encoding strategy delivered inconsistent and occasionally unsatisfactory results, especially for the Swiss Alps and Alaska, raising concerns about its potential overfitting.

Figure 2 provides a more detailed view of the classification results of the independent acquisition test data as derived with the region encoding strategy and bias optimisation as showing the best performance. For the Alps (Fig. 2a, b), the model demonstrated high accuracy in reconstructing the positions of glacier termini with small deviations from the reference data. For Southern Norway (Fig. 2c, d), the predictions approached near-perfection, largely due to the prevalence of clean ice in the region. For Alaska, it failed to accurately classify ice mélange and to identify the calving fronts (Fig. 2g), although the model successfully mapped a significant portion of debris-covered areas (Fig. 2e, f). For Southern Canada, the model exhibited robust performance in classifying debris-covered ice (Fig. 2i, we suspect errors in the reference data on the bottom side of i, and our model seems to outline the actual glacier terminus better), although some debris parts remained undetected (Fig. 2h).

The performance of the strategies exhibited a distinct trend as compared to the previous results based on the tile-based dataset. The independent acquisition tests present more challenging scenarios marked by larger domain shifts due to different sensors, imaging conditions and terrain features. Therefore, accuracy estimates derived from these tests are more reliable and better depict the actual expected performance of the strategies in global multitemporal applications.

We also utilised the independent acquisition test dataset for additional comparisons and validation. First, we compared the results derived with GlaViTU with a band ratio method. For this, we used a widely accepted band ratio method—Red/SWIR$_{1.6\mu m}$ > th$_1$ and Blue > th$_2$[17]. The optimal threshold values (th$_1$ = 3.2, th$_2$ = 0.17) were found by maximising IoU on the training data from the tile-based dataset. GlaViTU outperforms the band ratio, which is not capable of classifying debris-covered ice and produces many false positives for water bodies in all regions (Fig. 2 and Table 1). In Southern Norway, dominated by clean ice, the band ratio approaches the performance of GlaViTU (IoU = 0.915 vs. 0.933).

Moreover, we observed that GlaViTU yielded inconsistent results for small glaciers. To gain insights into its performance across glacier

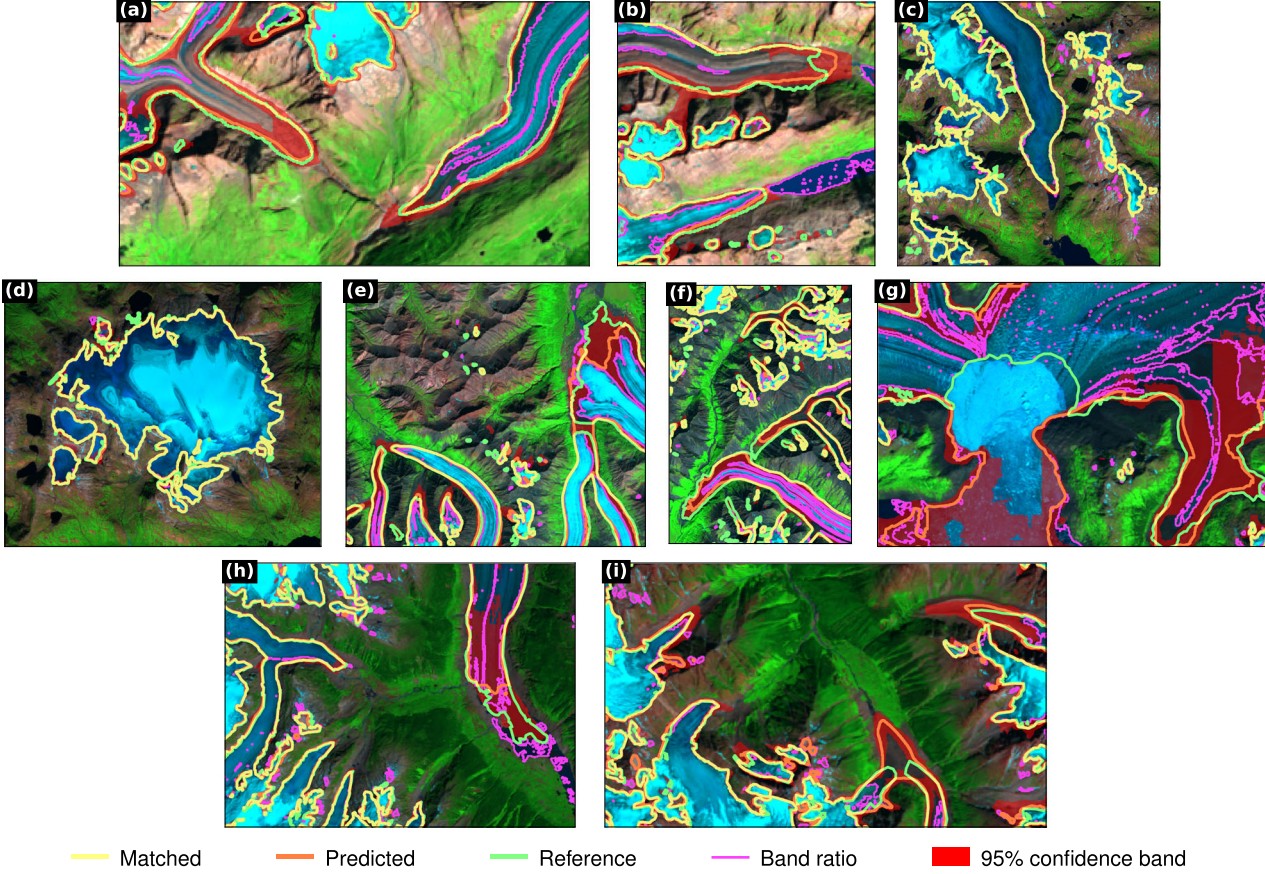

| — Matched | — Predicted | — Reference | — Band ratio | ■ 95% confidence band |

**Fig. 2 | Semantic segmentation results for the independent acquisition test data as derived using GlaViTU with regional encoding and bias optimisation. a**, **b** The Swiss Alps, **c**, **d** Southern Norway, **e**–**g** Alaska and **h**, **i** Southern Canada. The satellite images are presented in a false colour composition (R: SWIR$_{\approx 2.2 \mu m}$, G: NIR, B: R). Landsat images courtesy of the U.S. Geological Survey. Copernicus Sentinel data 2019.

scale variability, we evaluated its detection accuracy for different glacier sizes (Supplementary Table 5). The model showed high detection performance for glaciers >1 km$^2$ ($F_1$ > 0.9) and limited accuracy for glaciers <0.1 km$^2$ ($F_1$ < 0.4). We took 17 representative glaciers of varying sizes and debris cover from the independent acquisition test dataset and assessed the quality of the outlines in terms of area and distance deviations (Supplementary Table 6). The area deviation ranged from −10% to +10%, except Brattbreen (−11.90%) and Scimitar glacier (+16.21%), the former included a 0.0074 km$^2$ patch that was missed by the model and the latter missed a large fraction of debris-covered ice next to the lateral moraines in the reference; the median distance deviation approximately equalled the pixel size of the imaging sensor (10 m for Sentinel-2 and 30 m for Landsat) with 95th percentiles reaching 300 m, which is within the human expert uncertainty reported in the literature[46,47].

### Comparison of data tracks

We evaluated the impact of the three data tracks on the glacier mapping model performance across various subregions. For these tests, we used the global strategy. The results are summarised in Supplementary Table 7. Adding thermal data to the model did not provide substantial improvements and even led to decreased performance for about half of the subregions as compared to optical and DEM data alone (average IoU = 0.884 vs. 0.888 for the same subregions). In contrast, incorporating InSAR data consistently enhanced the model accuracy across all subregions where it was available (0.891 vs. 0.861), with the largest improvement for the Alps (0.873 vs. 0.844), the Southern Andes (SAN1, 0.890 vs. 0.874) and Scandinavia (SCA2, 0.909 vs. 0.836).

Supplementary Fig. 12 provides a qualitative illustration of the effects of adding the thermal band and highlights instances where adding thermal data harmed performance, particularly in the classification of debris-covered ice (Supplementary Fig. 12a–f), and led to increased false positives in ice mélange (Supplementary Fig. 12g). Conversely, Supplementary Fig. 7 demonstrates the advantages of incorporating InSAR data. Adding SAR backscatter and InSAR coherence improved the accuracy of glacier termini mapping (Supplementary Fig. 7a), enabled mapping of ice partially occluded by thin clouds (Supplementary Fig. 7b) and partially resolved the issues related to artefacts along coastlines (Supplementary Fig. 7c).

In summary, the evaluation revealed that in our experimental design, the addition of thermal data had limited benefits and, for about half the cases, even degraded model performance. The inclusion of InSAR data consistently enhanced the model performance across all tested regions.

### Uncertainty quantification

We leveraged Monte-Carlo dropout[44] and plain softmax scores to derive predicted class probabilities and used them to produce classification confidence estimates. Initially, when using Monte-Carlo dropout before predictive confidence calibration, we found that the model tended to exhibit significant underconfidence evidenced by high expected calibration error (ECE$_{100}$ = 0.81430). This finding, as illustrated in Fig. 3a, highlights the need for further calibration to enhance the reliability of these uncertainty estimates. Confidence calibration resulted in a remarkable reduction in the expected calibration error to ECE$_{100}$ = 0.00736. Figure 3b shows the improvement

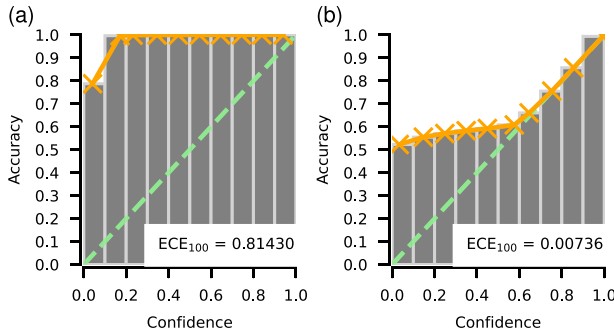

**Fig. 3 | Reliability diagrams for confidence derived with Monte-Carlo dropout. a** Before and **b** after predictive confidence calibration. ECE stands for expected calibration error, lower values indicate better calibration. Bins and orange curves depict the actual accuracy versus confidence as evaluated on the validation subset, and green lines show the ideal calibration case. After calibration, confidence aligns more closely with the actual accuracy, which enables interpreting predictive confidence in more absolute terms. In this particular case, one can expect that $X\%$ of the pixels predicted with confidence $X\% > 60\%$ are predicted correctly. Without this step, one can only compare confidence levels of predictions to each other. Source data are provided as a Source Data file.

after confidence calibration. Interestingly, we found that confidence estimates derived solely from plain softmax scores yielded outcomes almost identical to those obtained from Monte-Carlo dropout ($ECE_{100} = 0.80529$ before calibration and $ECE_{100} = 0.00660$ after). Moreover, the calibrated estimates derived from Monte-Carlo dropout and softmax scores were highly correlated, with the Pearson correlation coefficient lying between 0.873 and 0.884 (reported as the 2.5th and 97.5th percentiles obtained from bootstrapping). Hence, we suggest that confidence derived from softmax scores alone can replace Monte-Carlo dropout as a reliable means of estimating predictive uncertainty, providing lower computational costs required for inference.

Supplementary Figs. 3 and 6 provide visualisations of confidence rasters for representative tiles, offering insights into how these confidence estimates are distributed across different terrain features. Confidence tended to be consistently low for debris-covered ice no matter whether it was mapped correctly or not (Supplementary Figs. 3a–d and 6a–c). Confidence was also low for shadowed areas where the model failed to detect glacier pixels (Supplementary Fig. 6c). For correctly predicted ice mélange, our model assigned high confidence scores (Supplementary Fig. 3g). For misclassified ice mélange areas, confidence estimates exhibited variability, being low in certain settings (Supplementary Fig. 6d) and high in others (Supplementary Fig. 6e). We also reported 95%-confidence bands for the semantic segmentation results of the independent acquisition test data for generalisation tests in Fig. 2. The confidence bands exhibited similar patterns for debris and shadowed areas. In total, 83.97% of the misclassified pixels were included in the confidence bands, and $ECE_{100} = 0.00587$ on the independent acquisition test data. Notably, the misclassified parts of the debris-covered ice were mostly within the reported confidence bands, unlike ice mélange which was wrongly classified as glaciers with high confidence.

## Discussion
### Model performance
GlaViTU achieves high overall accuracy and often eliminates the need for manual corrections, offering a substantial improvement over well-established band ratio methods that form the backbone of the majority of semi-automated digitisation processes, including those used in GLIMS and RGI, as well as other deep learning models (see Supplementary Notes). We compared the results of GlaViTU against the human expert uncertainty presented in the literature,

notably the work by Paul et al.[46] and the GLIMS Analysis Comparison Experiments (GLACE) experiments[47]. Our analysis included a diverse subset of 17 glaciers, varying in size and debris cover, from the independent acquisition test dataset. The assessment was based on area and distance deviations, with most glaciers exhibiting deviations within ±10%, aligning closely with the uncertainties of human expert delineations. While the median distance deviations were on par with the pixel size of Sentinel-2 and Landsat imaging sensors, with 95th percentiles within the range of expert uncertainty, there are essential considerations to account for. The GLACE experiments and Paul et al. reported human uncertainties for mostly clean glaciers, excluding large debris-covered parts or extensive shadow areas which are significant sources of errors in inventories. Furthermore, studies such as these typically focus on a small selection of glaciers, which might not reflect the variety found in larger datasets, and they may give overly positive views of expert accuracy since the experts are aware their work is being compared against each other. In practical settings, manual mapping uncertainty is often much higher than reported in controlled studies. We have found discrepancies where expert uncertainty exceeded reported figures by a factor of five to ten, particularly in complex scenarios involving debris (e.g., seen in Fig. 2i and Supplementary Fig. 11d). These observations point to a substantial knowledge gap in manual mapping at scale and underscore the effectiveness of our automated method across diverse glacial landscapes while being scalable and fully reproducible, unlike human experts.

Challenges still persist, for instance, identifying debris-covered tongues remains a complex task due to their spectral similarity to surrounding rocks. GlaViTU also faces issues in accurately classifying shadowed ice and detecting small glaciers. The limited detection performance for smaller glaciers (<0.1 km²) can also be partially attributed to the uncertainties inherited in human-derived inventories, which arise from subjective decisions regarding the minimum glacier area to include, with typical values ranging from 0.01 km²[48] to 0.05 km²[49]. For dense ice mélange, our models exhibit a significant number of false positives because its spectral signature is similar to that of clean ice and it is underrepresented in our datasets. Furthermore, unexpected artefacts occasionally occur on coastlines, likely caused by similar-to-ice low water body reflectance in the shortwave-infrared range and zero slopes in the DEM for the masked ocean. It implicates the need for further model or algorithm refinement. For example, during model training, an adaptive sample strategy that feeds to the model more challenging samples could be beneficial to improve the performance on the difficult-to-classify targets. Alternatively, some of the targets could be treated in a special manner. For instance, calving fronts could be mapped with methods proposed by Wu et al.[50] or Heidler et al.[51]. Post-processing could be applied to eliminate some of the problems. For instance, sliver polygons on the medial moraines of glaciers could be filtered out based on shape descriptors, and artefacts at coastlines could be removed using masks that indicate glacier absence based on a priori knowledge. Also, the confidence bands present an opportunity to improve small glacier and debris cover mapping as discussed below.

### Multitemporal global-scale generalisation
Together with the model, we introduced five strategies to achieve global-scale glacier mapping. We evaluated these strategies in two different ways—with the tile-based test subset and with the independent acquisition test data that was compiled to test the generalisation. The outcomes of the two evaluations diverged. According to the tile-based evaluation, the regional and finetuning strategies are superior to others. Conversely, according to the evaluation based upon the data completely separated in either time or space, the region encoding strategy coupled with bias optimisation showed the best results, although the differences with the global strategy were minor.

Technically, the bias optimisation method can be implemented for any model. In our experimental design, however, it naturally complements the location encoding strategies that already contain a special block for bias introduction. The coordinate encoding strategy failed when classifying completely unseen data which may indicate its proneness to overfitting, further research might investigate regularisation methods for this strategy. As the second evaluation method is less biased, we assert that the accuracy estimates that we obtained during these temporal and spatial generalisation tests are more reliable in view of deploying the model to global multitemporal mapping, requiring generalisation in space, time and sensor characteristics. The first evaluation method can still be biased towards satellite sensors, imaging conditions or the procedures used to produce reference data (e.g., filtering out small snowpacks based on an area threshold). Notably, the tile-based evaluation is widely adopted[31,52,53], which may indicate overoptimistic model performances often reported in the literature, and thus, such evaluations should be approached with caution, particularly when seeking a model with high generalisation ability for large-scale operational applications. Still, other strategies can yield better results in some settings. For instance, regional models can perform exceptionally well when applied to the same sensor data and similar conditions as those present in the training dataset for a particular region. Overall, our models achieve accuracy and robustness high enough to observe significant decadal glacier area change as shown in our analysis detailed in Supplementary Notes. This analysis proves the capability of GlaViTU to monitor long-term changes in glacier extents and validates the model's utility in multitemporal studies. Furthermore, additional validation against a completely independent dataset, the Swiss Glacier Inventory 2016[54], provides further insights into the effectiveness and robustness of GlaViTU, as detailed in Supplementary Notes. This validation is crucial as it utilises data from sub-metre pixel resolution, considered the golden standard due to its higher accuracy, though it is not applicable globally and also includes subjective biases.

## Data tracks

The comparison of different data tracks within our glacier mapping framework provides valuable insights into the role of various remote sensing inputs in improving model performance. Firstly, our results suggest that the addition of thermal data, while initially promising due to its potential for distinguishing between surrounding bedrock and debris-covered ice[22,55], does not reliably enhance performance in our experiments. In fact, it leads to a decrease in accuracy in about half of the subregions. This outcome suggests that thermal data introduces complexities or uncertainties that our model struggles to adapt to, particularly in regions with debris cover. The thermal band, as a powerful predictor for clean ice and snow, may saturate the neural network during the early stages of training. Moreover, the lower spatial resolution of thermal bands (120 m for Landsat 5 TM, 60 m for Landsat 7 ETM+ and 100 m for Landsat 8 TIRS) can blur important features necessary for accurate glacier mapping, particularly in regions with debris-covered ice where more subtle details are needed to differentiate between debris and bedrock. Also, the value of thermal images decreases as the thickness of the debris layer approaches 0.5 m, effectively insulating the ice beneath and obscuring its thermal signature[56]. Therefore, the straightforward incorporation of thermal data may result in performance loss and should be approached with caution. In contrast, the positive impact of adding InSAR data across all subregions where it was available highlights its importance for glacier mapping. It improves overall accuracy and addresses specific challenges, such as accurate termini mapping or partial occlusion of ice with thin clouds. These improvements are likely due to the SAR's ability to penetrate clouds and InSAR's capacity to detect centimetre-scale deformations enabling more precise differentiation between ice and stable land. Consistent and global SAR data, however, are difficult to collect for the periods before 2015. Thus, it is reasonable to focus on developing further the Optical+DEM track to map glaciers before 2015 and the Optical+DEM+InSAR track after 2015 when Sentinel-1 data became globally available.

## Uncertainty quantification

Uncertainty quantification within our semantic segmentation models revealed some noteworthy findings. Before confidence calibration, the models exhibited a tendency towards underconfidence, which can be partially related to the use of label smoothing[57]. Also, we found plain softmax scores, which are computationally efficient and straightforward to obtain, yield confidence estimates comparable to those obtained from Monte-Carlo dropout, contrary to some reports in the literature. Indeed, several studies have highlighted the overconfidence of deep learning models and the limitations of using softmax scores as reliable uncertainty estimates[58]. Surprisingly, our findings indicated that plain softmax scores can be a valuable resource for estimating predictive uncertainty in our glacier mapping context.

The visualisations of predictive confidence offer insight into the distribution of confidence across different terrain features and provide additional context for refining our models and understanding their limitations. They can also serve as guidance for manual corrections, choosing the most informative samples to label within an active learning loop[59] or as a part of interactive semi-automated systems. Moreover, predicted confidence scores have the potential to be integrated into post-processing algorithms for further improving the accuracy of the mapping results. Additionally, the predictive confidence can be further linked with IoU as a measure of the model performance in the absence of reference data (see Supplementary Notes).

## Conclusions

In this paper, we presented GlaViTU, a hybrid convolutional-transformer model for glacier mapping from open multimodal satellite data. Also, we published a benchmark dataset for global-scale glacier mapping. GlaViTU consistently outperforms other deep learning models (see Supplementary Notes) across various regions, producing high-quality results for the majority of the images, approaching human expert uncertainty. Despite its successes, challenges persist, particularly in identifying debris-covered tongues, classifying shadowed ice and dealing with ice mélange. We introduced five strategies to achieve high generalisation for glacier mapping across regions and through time. Overall, the region encoding strategy coupled with bias optimisation is the best strategy allowing achieving IoU values above 0.85 for previously unobserved images on average (>0.75 for debris-rich and >0.9 for clean-ice regions). Our study reveals that both Monte-Carlo dropout and plain softmax scores can provide reliable confidence estimates for the model predictions after calibration, with plain softmax scores being a more computationally effective choice. These estimates can be instrumental for post-processing and understanding model limitations. In future research, the incorporation of explainable AI methods, e.g., integrated gradients[60], can further enhance model transparency and aid in addressing the remaining challenges. We proposed a relation for IoU estimation in the absence of reference data based upon predictive confidence, showcasing a strong correlation between estimated and actual local IoU values (see Supplementary Notes). Our work also includes an automated approach to derive ice divides from DEM as proposed by Kienholz et al.[61] (see Supplementary Notes), making an end-to-end workflow for the generation of glacier outlines.

This work also offers broader scientific insights into Earth observation and machine learning that go beyond glacier mapping. This paper underscores the importance of uncertainty quantification in model predictions, often overlooked in remote sensing and machine learning studies, providing a pathway for more transparent and interpretable AI applications in Earth sciences. The techniques introduced here can handle Earth observation data from different times and

sensors, are adaptable for studying various environmental phenomena and provide valuable tools for the Earth observation community to better monitor and address large-scale environmental changes. Outperforming other baseline models, we suggest the application of GlaViTU in various remote sensing domains, e.g., land-use and land-cover mapping and the monitoring of vegetation, forests, sea ice and ice sheets.

In conclusion, our study represents an advancement in the efforts towards fully automated global-scale glacier mapping, offering improved accuracy and efficiency, and enabling the generation of regularly updated global glacier inventories and long-term glacier area change analysis. These advances are key to improving the quality of downstream analyses, including, e.g., measurements and modelling of ice surface velocity, mass balance and glacier evolution. Challenges, however, remain, and further model and algorithm refinements are needed. We encourage the community to actively use our open-source code and benchmark datasets for future model development to help overcome these challenges. With the approaches and findings from this research, we are ready to launch a comprehensive, AI-based global glacier mapping initiative that will track glacier changes from the past thirty years and well into the future.

## Methods

### Study areas and datasets

We collected two datasets—a tile-based dataset that includes diverse glacier types across several regions worldwide and an independent acquisition test dataset that is specifically used to evaluate models on data spatially and temporally disjoint from the tile-based dataset as well as across different sensors and imaging conditions. The tile-based dataset combines publicly available optical, SAR and DEM data for 11 regions (23 subregions) worldwide—the European Alps (ALP), Antarctica (ANT), Alaska and Western America (AWA), Caucasus (CAU), the periphery of Greenland (GRL), High-Mountain Asia (HMA), low latitudes (TRP), New Zealand (NZL), the Southern Andes (SAN), Scandinavia (SCA) and Svalbard (SVAL). We designed the dataset to encompass a diverse spectrum of glaciers, including, e.g., clean-ice, debris-covered and vegetation-covered, situated in various surroundings such as alpine, polar and tropical as well as maritime and continental environments. This design choice allowed us to address the need for robust glacier mapping across various climates and terrains. The dataset covers approximately 9% or 19,000 of glaciers worldwide and represents about 7% of the glacierised area excluding the Greenland and Antarctic ice sheets. The earliest reference data are from the Antarctic region in 1988[62], while the most recent data are from Svalbard in 2020[43].

The tile-based dataset incorporates data from diverse sources. For optical data, we used top-of-atmosphere reflectance values of six bands, namely, blue, green, red, near-infrared and two shortwave-infrared bands (≈1.6 μm and ≈2.2 μm) from Landsat 5, 7, 8 and Sentinel-2. As SAR features, $\sigma_0$-calibrated amplitude images acquired by ENVISAT and Sentinel-1 from both ascending and descending orbital paths were employed. In addition to the amplitude, interferometric SAR (InSAR) coherence images were used if available. Shuttle Radar Topography Mission DEM (SRTM), ALOS World 3D 30m (AW3D30) and Copernicus GLO-30 DEM (Cop30DEM) were used to represent elevation data and terrain slopes. We employed regional inventories for glacier outlines in the Alps[40], New Zealand[41], Northern Scandinavia[42] and Svalbard[43] (we received a preliminary version of the dataset from the authors), while GLIMS[16] directly provided reference data for the remaining regions, with only a few original references found in the metadata[15,62–64]. Optical imagery was chosen to match the exact dates specified in glacier inventory metadata, particularly focusing on late summer acquisitions at the end of the ablation season before the first major snow event to ensure minimal seasonal snow

cover. For SAR and DEM data, where exact date matching was not feasible, we selected images within a constrained temporal window—up to a month for SAR and seven years for DEM, except Antarctica and Greenland which are not covered by SRTM and where we used AW3D30. All images were resampled to 10 m resolution to match with the highest spatial resolution of Sentinel-2, nearest neighbour interpolation was applied to the optical bands, while bilinear interpolation was used for resampling the rest of the data. Following a widely accepted practice in remote sensing studies, we divided all regions into near-squared tiles of an approximate size of $10 \times 10$ km², and randomly selected approximately 60% tiles for training, 20% for validation and 20% for testing. Figure 1 shows an overview of the tile-based dataset and study areas, and Table 2 provides its summary.

To deal with the varying feature sets across different subregions, we identified three distinct data tracks for use in the subsequent experiments. These tracks are as follows:

1. Optical+DEM: includes all subregions, incorporating six optical bands as described above and DEM data including two channels of stacked elevation and slope.
2. Optical+DEM+thermal: includes all subregions where thermal data (one channel) from the Landsat satellites are accessible, in addition to the optical and DEM data identical to the previous track.
3. Optical+DEM+InSAR: includes all subregions where InSAR coherence data from Sentinel-1 satellites are available. Also, co-pol $\sigma_0$ values were included in this track, resulting in a total of four inputs. Optical and DEM data were the same as in the first track. The InSAR coherence input contained two channels of stacked InSAR coherence maps, and the co-pol $\sigma_0$ input included two channels of stacked backscattering images from ascending and descending orbital paths.

The independent acquisition test dataset comprises data from GLIMS[16] to assess the generalisation of the models to new acquisitions in different temporal and spatial contexts. Incorporating these independent acquisition test data provides insights into mapping performance across diverse regions and temporal scales not present in the tile-based dataset, thus aligning with our long-term goal of automating global and multitemporal glacier mapping with different imaging sensors. To evaluate temporal generalisation, we obtained data from the Swiss Alps (259 glaciers covering 336.74 km²), a subset of ALP, from 1998, and from Southern Scandinavia (1508 glaciers, 839.07 km²), the same area as SCA1, from 2019[42]. The Swiss Alps are characterised by debris-covered tongues and ice in shadow, while Southern Scandinavia predominantly features clean ice. The temporal generalisation tests also served as cross-sensor and cross-DEM generalisation tests as both the optical sensors and DEMs used in these tests were completely different from those in the tile-based dataset. The tile-based dataset consists of Sentinel-2 imagery and Cop30DEM tiles for the Swiss Alps, while additional data rely on Landsat 5 and SRTM. Similarly, SCA1 from the tile-based dataset utilised Landsat 5 and AW3D30, whereas we employed Sentinel-2 and Cop30DEM for the test. For spatial generalisation tests, we selected an area in Alaska (1108 glaciers, 5064.35 km², 2009, Landsat 5 and AW3D30) and an area in Southern Canada (980 glaciers, 2297.90 km², 2005, Landsat 5 and AW3D30). Both areas are characterised by glacier tongues with prominent debris cover. Furthermore, the Alaska region includes water-terminating glaciers with ice mélange, presenting a considerable challenge for accurate mapping. These independent acquisition test data were only used to evaluate the models trained on the training tiles from the tile-based dataset and were not utilised for training. Supplementary Fig. 13 shows the additional areas for temporal and spatial generalisation tests.

**Table 2 | Tile-based dataset summary**

| Region | | Subregion | Year | Number of tiles | | | Features | | | | | | Glacier area, km² | Number of glaciers | Debris coverage, % | Reference |
|---|---|---|---|---|---|---|---|---|---|---|---|---|---|---|---|---|
| | | | | Train | Val | Test | Optical | DEM | Thermal | Co-pol σ₀ | Cross-pol σ₀ | InSAR coherence | | | | |
| Alps | ALP | ALP | 2015 | 177 | 59 | 60 | ✓ | ✓ | × | ✓ | ✓ | ✓ | 1317.40 | 3217 | 22.93 | 40 |
| Antarctic | ANT | ANT1 | 1988 | 120 | 40 | 40 | ✓ | ✓ | ✓ | × | × | × | 6144.66 | 175 | n/a | 62 |
| | | ANT2 | 2001 | 81 | 27 | 28 | ✓ | ✓ | ✓ | × | × | × | 1946.21 | 253 | n/a | GLIMS |
| Alaska and Western America | AWA | AWA1 | 2005 | 95 | 32 | 32 | ✓ | ✓ | ✓ | × | × | × | 7482.72 | 595 | 30.79 | GLIMS, RGI |
| | | AWA2 | 2010 | 99 | 33 | 33 | ✓ | ✓ | ✓ | × | × | × | 12657.47 | 414 | 24.97 | GLIMS, RGI |
| Caucasus | CAU | CAU | 2014 | 22 | 7 | 8 | ✓ | ✓ | ✓ | × | × | × | 571.61 | 524 | 18.12 | 63 |
| Greenland | GRL | GRL1 | 1999 | 70 | 24 | 24 | ✓ | ✓ | ✓ | × | × | × | 3797.79 | 244 | 5.95 | GLIMS, RGI |
| | | GRL2 | 2000 | 108 | 36 | 36 | ✓ | ✓ | ✓ | × | × | × | 3021.12 | 551 | 5.34 | GLIMS, RGI |
| High-Mountain Asia | HMA | HMA1 | 2005 | 33 | 11 | 11 | ✓ | ✓ | ✓ | ✓ | × | × | 1022.03 | 418 | 57.65 | GLIMS |
| | | HMA2 | 2007 | 53 | 18 | 18 | ✓ | ✓ | ✓ | × | × | × | 1437.97 | 1252 | 34.62 | GLIMS |
| | | HMA3 | 2009 | 33 | 11 | 11 | ✓ | ✓ | ✓ | ✓ | × | × | 2295.91 | 950 | 15.02 | GLIMS, RGI |
| | | HMA4 | 2007 | 16 | 5 | 6 | ✓ | ✓ | ✓ | × | × | × | 892.62 | 469 | 12.51 | GLIMS, RGI |
| | | HMA5 | 2008 | 30 | 10 | 11 | ✓ | ✓ | ✓ | × | × | × | 767.29 | 850 | 45.29 | GLIMS |
| Low Latitudes | TRP | TRP1 | 2015 | 0 | 0 | 1 | ✓ | ✓ | ✓ | ✓ | × | ✓ | 0.56 | 6 | 0.00 | GLIMS |
| | | TRP2 | 1998 | 22 | 7 | 8 | ✓ | ✓ | ✓ | × | × | × | 502.34 | 681 | 5.46 | GLIMS |
| New Zealand | NZL | NZL | 2019 | 92 | 31 | 31 | ✓ | ✓ | ✓ | ✓ | ✓ | ✓ | 652.44 | 2891 | 32.91 | 41 |
| Southern Andes | SAN | SAN1 | 2016 | 58 | 19 | 20 | ✓ | ✓ | ✓ | ✓ | × | ✓ | 541.49 | 1103 | 8.30 | GLIMS |
| | | SAN2 | 2016 | 207 | 69 | 70 | ✓ | ✓ | ✓ | ✓ | × | × | 7375.86 | 3172 | 6.21 | GLIMS |
| Scandinavia | SCA | SCA1 | 2006 | 81 | 27 | 27 | ✓ | ✓ | ✓ | × | × | × | 872.35 | 689 | 4.26 | 64 |
| | | SCA2 | 2018 | 13 | 5 | 5 | ✓ | ✓ | ✓ | ✓ | ✓ | ✓ | 54.75 | 127 | 0.57 | GLIMS |
| Svalbard | SVAL | SVAL1 | 2020 | 120 | 32 | 33 | ✓ | ✓ | × | ✓ | ✓ | ✓ | 2782.39 | 207 | 14.43 | 43 |
| | | SVAL2 | 2020 | 68 | 31 | 30 | ✓ | ✓ | × | ✓ | ✓ | ✓ | 599.32 | 278 | 1.26 | 43 |
| | | SVAL3 | 2020 | 38 | 13 | 13 | ✓ | ✓ | × | ✓ | ✓ | ✓ | 850.49 | 98 | 0.00 | 43 |
| | | | Total: | 1636 | 547 | 556 | | | | | | | 57,586.80 | 19,164 | | |

Debris coverage is adapted from Herreid and Pellicciotti[79].

## GlaViTU

We extended our previous work[65], where we proposed a preliminary version of GlaViTU (Supplementary Fig. 10), a hybrid of a convolutional network and a transformer for large-scale glacier mapping. It incorporates simplified building blocks from SEgmentation TRansformer (SETR) with progressive upsampling (PUP) decoder[34] and U-Net[32] with residual connections[66]. By sequentially placing the transformer and convolutional subnets, GlaViTU ensures that the global image context is learnt initially followed by fine-grained feature extraction using the convolutional subnet. This design choice is motivated by the observation that purely transformer-based models face challenges in making detailed dense predictions[35]. Combining the strengths of convolutional and transformer models, GlaViTU effectively captures both local-level and global-level features in multi-source heterogeneous remotely sensed data enabling accurate glacier inventory production and better generalisation. This paper introduces a modified version of GlaViTU with an enhanced data fusion block and a more advanced training routine. Supplementary Fig. 8 shows the modified data fusion block for the case of three inputs. Inspired by Hu et al.[67], we added one squeeze-and-excitation block in each convolutional branch to incorporate more cross-channel interactions and add feature weighting before fusing the multi-source inputs.

The models were trained with image patches of size 384 × 384 pixels randomly extracted from the training subset tiles. We employed the Adam optimiser[68] to find the model parameters by minimising the focal loss with the focal parameter set to 2[69]. We added a deep supervision branch with an identical loss function right after the transformer subnet (Supplementary Fig. 10). To further enhance the training process, we applied label smoothing with the smoothing parameter of 0.1[57,70]. To dynamically adjust the learning rate, we utilised a cosine decay schedule with warm restarts as proposed by Loshchilov and Hutter[71] to avoid local minima and extensive plateau areas in the loss landscape. The initial learning rate is $5e^{-4}$, training was divided into four cycles, each consisting of 10, 20, 40, and 80 epochs, respectively. After each cycle, the learning rate was set back to its initial value. At the end of each epoch, the models were evaluated on the validation set. Only the models that demonstrated the best performance on the validation set were saved and selected for further evaluation and analysis. To augment the training data and improve the generalisation, on-the-fly data augmentation techniques were applied. These techniques included random vertical and horizontal flips, rotations, cropping and rescaling of the image patches. Additionally, we employed random contrast adjustments, both channel- and pixel-wise Gaussian noise and feature occlusion, which is dropping out rectangular regions from the inputs during training.

## Strategies towards global glacier mapping

To achieve global glacier mapping, we explored different strategies that involve both traditional training methods and location encoding techniques. We present the following five strategies:

1. Global: a single model was trained using data from all regions collectively. The model learnt to generalise across diverse regions, capturing global patterns and characteristics of glaciers.
2. Regional: individual models were trained for each region. This allows capturing region-specific features and nuances, potentially leading to improved performance in each specific area. It is also possible to train one model for a cluster of regions. This can be particularly useful for obtaining models applicable to unseen regions by clustering regions in the dataset that share common features with an unseen area of interest.
3. Finetuning: the pretrained global model was finetuned for each region using data from this specific region. By doing so, we aimed to achieve a balance between capturing globally common patterns and incorporating region-specific information. Similarly to the previous strategy, the global model can be finetuned to a

cluster of regions. For finetuning, we used only the last cycle (80 epochs) of the training process with a decreased learning rate of $5e^{-5}$.

4. Region encoding: we encoded each region as a one-hot vector and fed it to the models. We assigned a separate position in the vector for each region, including an additional position for the 'unseen' region, resulting in a total of 12 positions. During training, random samples labelled as 'unseen' were introduced to enable the model to learn a generalised representation for regions not present in the dataset.

   There is, however, a risk of learning biases associated with not only the regional terrain features but also with the imaging sensor such as spatial resolution and spectral response and the atmospheric conditions such as cloud cover, aerosol content and illumination conditions in the images from the training dataset. These biases can potentially limit the generalisability of the region encoding to new images if applied as is. To overcome this risk, we propose inference-time bias optimisation. It aims to reduce the domain shift by finetuning the region vector so it includes soft labels of several regions adjointly and, thus encoded biases present in the whole training dataset. As the criterion, we minimised the predicted uncertainty under the assumption that it is related to the model performance. Bias optimisation is formulated as:

$$-\sum_{i=1}^{N} \mathbf{p_i}(\mathbf{r}) \log_C(\mathbf{p_i}(\mathbf{r})) \to \min_{\mathbf{r}}, \qquad (1)$$

$$\mathbf{p_i}(\mathbf{r}) = f\left(\mathbf{I}_i, \exp(\mathbf{r})/\sum_{j=1}^{12} \exp(r_j)\right), \qquad (2)$$

where $\mathbf{r}$ is the vector of optimised biases, $\mathbf{p_i}(\mathbf{r})$ is the vector of predicted probabilities for sample $i$, $f$ is the mapping model, $\mathbf{I}_i$ is the test image features, $\exp(\mathbf{r})/\sum_{j=1}^{12} \exp(r_j)$ is the region vector, $N$ and $C$ are the number of samples and the number of classes, respectively, and $\log_C()$ denotes the logarithm base $C$. The finetuned region vectors were fed into the mapping model for a regular inference instead of the one-hot region vectors. We solved this optimisation problem for a whole scene with the Adam optimiser[68] and a constant learning rate ($1e^{-4}$) for 100 epochs or until convergence.

5. Coordinate encoding: we encoded latitude ($\lambda$) and longitude ($\phi$) of a patch centroid were encoded as a four-dimensional vector: ($\sin(\lambda), \cos(\lambda), \sin(\phi), \cos(\phi)$). Such encoding naturally solves the problems of data normalisation and discontinuity around the 180th meridian. By incorporating coordinate information directly into the networks, it was expected that the models could learn spatial relationships and region-specific features in the patches based on their geographical locations.

Supplementary Fig. 9 illustrates the modified fusion block utilised in the region and coordinate encoding strategies, which enabled the integration of location information into the network architecture. Prior to the input fusion, the location vectors were processed with a simple feed-forward network. After that, they were added to the low-level features extracted from the images introducing location-specific biases. The fusion block was jointly trained with the rest of the model.

## Uncertainty quantification

Uncertainty quantification for dense prediction tasks has been tackled with Bayesian neural networks[72] or their approximations[73], deterministic methods[74,75], model ensembling[74] and other methods[76]. However, most studies neglect uncertainty calibration that potentially impacts the reliability and interpretability of uncertainty estimates.

This study introduces uncertainty calibration in the context of glacier mapping.

A necessary step in assessing mapping uncertainty is an accurate estimation of class probabilities. To do so, we used two methods:

- Monte-Carlo dropout[44] that applies dropout during both training and inference in deep neural networks. By performing multiple forward passes with dropout for each sample, we get a distribution of predictions, allowing us to obtain potentially more reliable probability estimates. The class probabilities are obtained as the average softmax scores among all forward passes.
- Plain softmax scores that are outputs of the last layer with applied softmax activation[45] from one forward pass. In general, softmax scores are not equal to probabilities as they tend to provide overconfident results[76,77]. Nevertheless, we investigated whether plain softmax scores provide comparable results to Monte-Carlo dropout. If they do, this presents a significant advantage by substantially speeding up the inference process.

As a measure of confidence, we employed the Shannon-entropy–based metric:

$$\text{conf}\,(\mathbf{p}) = 1 + \sum_{i=1}^{C} p_i \log_C (p_i), \tag{3}$$

where $\mathbf{p} = (p_1, ..., p_C)$ is the class probability vector, $C$ is the number of classes, and $\log_C()$ denotes the logarithm base $C$.

Estimated confidences may misalign with the actual accuracies. Therefore, it is important to calibrate the derived confidence to ensure accurate and meaningful uncertainty estimates. To do so, we searched for a confidence calibration model $R: [0, 1] \rightarrow [0, 1]$ that satisfies the following expression as closely as possible for $\forall\, \gamma \in [0, 1]$ as $N \rightarrow \infty$[77]:

$$\frac{\sum_{i=1}^{N} \mathbb{1}\{\text{argmax}\,(\mathbf{y_i}) = \text{argmax}\,(\hat{\mathbf{y}}_i)\} \cdot \mathbb{1}\{R[\text{conf}\,(\hat{\mathbf{y}}_i)] = \gamma\}}{\sum_{i=1}^{N} \mathbb{1}\{R[\text{conf}\,(\hat{\mathbf{y}}_i)] = \gamma\}} \rightarrow \gamma, \tag{4}$$

where $\gamma$ is a confidence, $N$ is the number of samples, $\mathbf{y_i}$ is a one-hot-encoded reference label, $\hat{\mathbf{y}}_i$ is predicted class probabilities, and $\mathbb{1}\{\}$ stands for the indicator function. The left side of Equation 4 represents the overall model accuracy for the predictions with the confidence level of $\gamma$, and the purpose of model calibration is to align the predicted confidence, $\text{conf}\,(\hat{\mathbf{y}}_i)$, with this accuracy. We modelled $R$ with kernel ridge regression and fitted it on the confidence estimates and accuracies derived from the validation set.

The quality of the confidence estimates was evaluated with reliability diagrams, which depict bins of the predicted confidence versus the actual accuracy within these bins, and the expected calibration error given as:

$$\text{ECE}_B = \sum_{i=1}^{B} b_i \cdot |\text{Accuracy}_i - \overline{\overline{\gamma}}_i|, \tag{5}$$

where $B$ is the number of equally spaced confidence bins, Accuracy$_i$ is the overall accuracy in bin $i$, $\overline{\overline{\gamma}}_i$ is the mean predicted confidence in bin $i$, and $b_i$ is the fraction of samples in bin $i$.

### Accuracy assessment

We assessed the classification performance using IoU as a pixel-wise metric:

$$\text{IoU} = |T \cap P| / |T \cup P|, \tag{6}$$

where $T$ and $P$ are the reference and predicted glacier pixels, respectively.

To assess the detection performance, we utilised precision, recall and F$_1$ score given as:

$$\text{Precision} = |T \cap P| / |P|, \tag{7}$$

$$\text{Recall} = |T \cap P| / |T|, \tag{8}$$

$$\text{F}_1 = 2 \cdot |T \cap P| / (|T| + |P|), \tag{9}$$

where $T$ and $P$ are the sets of the reference and detected glaciers, respectively. A predicted glacier was considered to match with a reference one if their intersection area consisted of more than 50% of both of them individually.

The area deviations were evaluated in absolute and relative terms as:

$$\Delta A = A_{\text{pred}} - A_{\text{ref}}, \quad \delta A = \left( A_{\text{pred}} - A_{\text{ref}} \right) / A_{\text{ref}}, \tag{10}$$

where $A_{\text{pred}}$ is a predicted glacier area, and $A_{\text{ref}}$ is a reference area. The distance deviations were estimated using the PoLiS metric[78] that quantifies the dissimilarity between two polygons by evaluating the average distance from each node of one polygon to the closest point on the boundary of the other and vice versa:

$$\overline{\rho(\text{pred}, \text{ref})} = \frac{1}{|\{p \in \text{pred} \cup \text{ref}\}|} \left( \sum_{\{p \in \text{pred}\}} \rho(p, \text{ref}) + \sum_{\{p \in \text{ref}\}} \rho(p, \text{pred}) \right), \tag{11}$$

where $p$ is a point of a boundary, ref and pred are the reference and predicted boundaries, respectively, and $\rho$ denotes the Euclidean distance. We sampled polygon nodes $p$ every 10 m along the boundaries to calculate this metric. We also reported median and 95th percentile values in addition to mean values to provide a more detailed assessment of the distance deviation distribution.

### Computational resources

We trained and deployed the models on cloud servers equipped with NVIDIA RTXA6000 GPUs, 16-core 2.3 GHz CPUs and 110 GB RAM. Training one model on the tile-based dataset took approximately two weeks, with hard drive input-output operations likely being the key performance bottleneck. Regional model training ranged from one to two days depending on the sample size. Fine-tuning a model took from a few hours to a day. Applying the models to the independent acquisition test data for the generalisation tests typically took only a couple of minutes. When using bias optimisation, it could take several minutes to 2 h on top, depending on the size of the area of interest.

## Data availability

The data collected and analysed in this study have been deposited in the NIRD database under accession code https://doi.org/10.11582/2024.00168. Source data are provided with this paper.

## Code availability

Our codebase and the pretrained models are available at https://github.com/konstantin-a-maslov/scalable_glacier_mapping/tree/v1.0.

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

## Acknowledgements

This research was financed by the Research Council of Norway under the "Researcher Project for Scientific Renewal" (project MASSIVE, no. 315971) awarded to T.S., C.P., A.S. and K.A.M. Additional support was provided by Open Clouds for Research Environments under the EU H2020 programme (project MATS_CLOUD, no. 824079) for access to the CREODIAS cloud computing platform, awarded to C.P., T.S. and K.A.M.

## Author contributions

K.A.M., C.P. and T.S. designed the study. K.A.M. implemented the methods and conducted the analysis. K.A.M., C.P., T.S. and A.S. discussed the results extensively. T.S. had the project idea and led the project with C.P. K.A.M. wrote the manuscript. C.P., T.S. and A.S. reviewed the manuscript.

## Competing interests

The authors declare no competing interests.
