## [Transparent Peer Review file · Nature Communications]

Globally Scalable Glacier Mapping by Deep Learning Matches Expert Delineation Accuracy

Corresponding Author: Mr Konstantin Maslov

Version 0:

Reviewer comments:

Reviewer #1

(Remarks to the Author)
Please see attached file.

(Remarks on code availability)
Please see attached file.

Reviewer #2

(Remarks to the Author)
General comments

Automated glaciers classification is a challenging task but also an important and valuable one as glaciers integrate climate over decadal timescales and are therefore useful indicators of trends rather than variability.

This is an impressive study in terms of its comprehensive nature and extensive analysis covering a wide range of glaciated regions, model design and tuning, and image types/sensors. It is certainly worthy of publication and I have no substantive comments regarding the methods or findings.

However, as presented, I do not believe it is appropriate for Nat Comms which is a broad readership journal covering all disciplines. In its current form it is more appropriate for an IEEE ML or remote sensing journal such as TGARRS, TIP or ISPRS. The m/s is replete with ML related jargon, often not explained. Terms such as Monte Carol Dropout, SoftMax and loU (not spelled out) are common to the ML community but far less so to the wider science community. It is not just the terminology used though. The heavy emphasis on comparison to existing ML approaches and performance metrics is common practice in IEEE publications but less appropriate to Nature family journals. There is frequent reference to supplementary figures and material because these are relevant to these performance criteria. The writing style and focus of the m/s is far more in keeping with such more technical journals and it would fit well there and would be a valuable contribution.

Specific comments

The paper goes from introduction straight to results bypassing data sets altogether, which does not appear until the methods. Crucially there is no discussion anywhere in the main text what the source of the ground truth was and this appears to be manually derived inventories from different sources acquired at different dates. It is unclear if the authors matched the imagery used to the dates of the inventories and with what latitude. This is important because some of those inventories likely cover more than one year. How long a time gaps is acceptable?

Related to this is the complete lack of any discussion of the timing of the imagery which is crucial for reliable classification. Late summer is needed to reduce the effects of variable seasonal snowcover that is difficult to impossible to differentiate from glacier if present. This issue is not discussed at all, which I found surprising. Despite no mention of this, the results are

excellent but it is unclear if this is because of careful choice of image data/season or not?

(Remarks on code availability)

Looks comprehensive and well documented but I did not try to run it myself

Reviewer #3

(Remarks to the Author)

Review of Maslov et al., Towards Global Glacier Mapping with Deep Learning and Open Earth Observation Data, submitted to Nature Communications

by Romain Hugonnet, University of Washington

Statement of expertise:

I have expert knowledge in:

- Remote sensing data,
- Glacier delineation techniques,
- Spatial uncertainty analysis.

Although I have machine learning knowledge, I lack advanced knowledge on convolutional-transform deep learning models, and recommend that at least another reviewer has that expertise.

General comment:

The authors present a very interesting study utilizing deep learning for glacier outline delineation, tested under varied calibration schemes, imagery types, spatial and temporal scales, and glacier surface conditions worldwide. The paper focuses largely on methodological development, improvement and optimization of the methodology, and the release of a related dataset for 9% of glaciers.

While the study is presenting very valuable work and is well-written, I am not sure if it fits for this type of journal, for the simple reason that it is very methods-focused and thus hard to read/access for a wider audience not specialized in machine-learning. It might be a better fit for a methods journal.

In terms of impact, I think that either an application extension using this method for all glaciers globally, or a methods comparison evaluating against traditional mapping methods in detail would allow to draw clearer conclusions for a larger community.

In this study, the developments are valuable but main conclusions somewhat limited:

- In terms of glaciological knowledge, we do not yet learn anything from newly derived datasets (limited to 9% of glaciers, and the authors do not analyse the area changes that can be estimated from the multi-temporal outlines, they are probably keeping this for later scaling efforts),
- In terms of methodological development for glacier delineation, while the methods are showing very promising results for large-scale automation and we learn a lot from comparing different calibration/data/scales, the authors do not reach substantial conclusions on the improvement compared to current semi-automated methods and published datasets.

In short, the study currently does not help answering any of the two following main questions it relates to:

- (Glaciological focus) How much has glacier area changed in the past decades (regional changes, etc)?
- (Methodological focus) Can we map glaciers with deep learning better, or with equal confidence, than with previously semi-automated (=manually-corrected) delineation methods and existing datasets?

My main criticism is of course with the second question (the first one being a choice of the authors to not scale yet and focus on methods), for which I have some detailed comments below.

Main comment 1: Reference data to evaluate accuracy and precision more reliably?

The authors point out in their main text (and more extensively in their supplementary materials) that there is an absence of reference data, which they circumvent in the rest of the study by implementing a confidence-based IoU estimation. The approach is interesting but, from that point on, it is somewhat hard to assess how much the confidence values can really be trusted, and most importantly how the results might compare to semi-automated mapping (which could be fairly easy to run against the deep learning approach).

There are several regions where glacier outlining is done at very high resolution based on local aerial datasets and extensively checked manually (golden example: the Swiss Alps, e.g. GLAMOS and Linsbauer (2021)), which could be used as "reference" to compare to temporally-close estimates from your method, and compare against more classical methods.

Alternatively, the authors could manually delineate/adjust a few dozens glaciers (or even hundreds, as done by some terminus mapping studies over Greenland/Antarctica) across the wide range of surface conditions and acquisition dates that they want to test, to create their own reference dataset, possibly using local datasets of higher resolution to ensure limited manual errors.

A reference dataset, even limited, would be invaluable to be more conclusive when stating how good an improvement the data learning approach makes compared to traditional methods, and what we can expect from scaling.

Main comment 2: Spatial considerations for IoU metric

It is currently unclear how the IoU metric is aggregated in space. Errors in delineation are typically a function of the instrument resolution, but not the glacier size, so they shrink with increasing glacier size when measured in percentage (what the IoU does per-glacier).

The authors report a much better performance in regions with large glaciers (Antarctica, Svalbard, Andes, Svalbard), and a lesser performance for regions with smaller glaciers, which lets me think that maybe the IoU was not area-weighted per-glacier when reporting regional-scale values?

In any case, a dependency on glacier area likely exists and should be accounted in the analysis of the results, as well as a other variables of interest (%age of debris cover?) to discuss the result.

Standardizing depending on the glacier area would also allow metrics to be comparable between different regions.

Line-by-line comments:

L8: "while" doesn't fit this sentence, "and"?

L19-23: Careful not to oversell the importance of glaciers for freshwater, it is limited and often exaggerated in introductions, see Gascoin (2023).

L24-26: GLOFs are actually decreasing in magnitude, see for example Veh et al. (2023).

L122: Define Intersection over Union acronym

L335/Fig.3: Define Expected Calibration Error acronym.

The text is very well written, but I think the authors would benefit from adapting some of the language they use for a broader audience of scientists in cryospheric sciences, or clearly defining technical terms early in the text.

References:

Linsbauer, A., Huss, M., Hodel, E., Bauder, A., Fischer, M., Weidmann, Y., Bärtschi, H. & Schmassmann, E. 2021, The new Swiss Glacier Inventory SGI2016: From a topographical to a glaciological dataset. *Frontiers in Earth Science*, 22, doi:10.3389/feart.2021.704189.

Gascoin, S. (2023). A call for an accurate presentation of glaciers as water resources. *WIREs. Water*.
<https://doi.org/10.1002/wat2.1705>

Veh, G., Lützw, N., Tamm, J., Luna, L. V., Hugonnet, R., Vogel, K., Geertsema, M., Clague, J. J., & Korup, O. (2023). Less extreme and earlier outbursts of ice-dammed lakes since 1900. *Nature*, 1–7.

(Remarks on code availability)

The repository is not fully ready but already in very good state: detailed README, data links, install guidelines, hardware notes, script descriptions, function modules nicely sorted.

Version 1:

Reviewer comments:

Reviewer #1

(Remarks to the Author)

Please find my comments in the attached review file.

(Remarks on code availability)

Please find my comments regarding the repository in the attached review file.

Reviewer #3

(Remarks to the Author)

Review of "Towards Global Glacier Mapping with Deep Learning and Open Earth Observation Data" by Maslov et al., round 2

by Romain Hugonnet, University of Washington

General comment

The authors have improved their manuscript to be more accessible to a broad readership, moving several methodological elements containing machine-learning jargon to the Methods section, and renaming some "own-coined" terminology for clarity based on comments also echoed by other referees. The quality of both text and figures is still high, and now the manuscript is also more generally accessible.

Additionally, the authors have partly addressed my main comment about investigating quantitatively if their automated

prediction is on par with human expert delineation, which I think has greatly helped re-focused the study's conclusions into a clear applicative result in terms of mapping accuracy.

However, the authors did not address the core of that comment, that is that GLIMS outlines cannot truly serve as a reference for estimating accuracy, and overlooked some of my propositions to remedy that (that seem to have been misunderstood). I think that this limitation is still a strong one, but could be fairly simply remedied by gathering a small dataset of very-high resolution outlines (and only outlines) that exist in different regions with local aerial surveys, which are small datasets that authors are often willing to share. Those would be largely error-free compared to GLIMS, and would make the perfect dataset (even if a small sample) for robust "validation", to fully back up the statement now made by the authors that their predictions are "on par" with human expert delineation.

1/ Short comment on claims on impact in rebuttal

The authors repeatedly insist in their rebuttal to not forget that the advances from their methods are mostly that mapped outlines are consistent and scalable without the use of many human resources (as they are automated), which is in itself a great value. While of this is of course true, based on the state of the field, a new glacier outlines dataset would likely have limited scientific impact if its accuracy is not rigorously proven to match existing human-based methods. And this, even if it provides new multi-temporal outlines with less effort.

The reasons being, for instance, that:

- Current glacier mass change estimates are not hampered that much by errors in modern glacier outlines due to temporal differences, as elevation change on bare-rock near the margin of the glacier (or where the glacier has retreated during a period) is nearly zero, so errors due to delineation are actually minimal (and knowingly over-estimated in the mass change community by applying a large % of error); Errors in delineation do affect specific mass change rates a lot (= mass change divided per the glacier area) but those are usually only used to interpret the climatic signal, and not for providing accurate estimation of direct volume/mass changes used for sea-level rise, river runoff, etc,
- Same reasoning for velocity estimates, where measured velocities near glaciers (or where glaciers have retreated) are close to zero, and so delineation errors also have a minimal impact on velocity errors, even if they are not from the right time period.

In short, the accuracy of the derived outlines remains the most important criteria to hope supersede current inventories with GlaViTa, and remains definitely more important than the perspective to produce multi-temporal outlines thanks to the scalability of the method.

In regards to the above, I think several statements are currently a bit over-stated by the authors in the introduced paragraphs at lines 58-80 and need to be revised.

2/ Main comment: adding an error-free validation dataset from "ground truth" data

In my opinion, the biggest limitation of this study remains the same, which is that the test dataset is a subset of GLIMS outlines that themselves have large errors, and so, despite the large sample tested, it is hard to truly assess the accuracy of the algorithm and interpret its accuracy.

As the authors state themselves in their rebuttal: "However, as we only compare our results with manually derived inventories, we cannot claim any improvement by the model over them in terms of accuracy."

In response to my previous review, the authors introduced two new evaluation exercises:

1. Comparing GlaViTa to simple band-ratio methods,
2. Evaluating GlaViTa errors with new metrics in a subset of the test dataset. They use instead the statistical mean and spread of distance deviation instead of the IoU, which I also believe are much better suited to evaluating the structure of errors in this type of predictions and are a great addition.

Those are definitely valuable, but do not address the above issue.

My recommendation (that seems to have been misunderstood the first time) to address this issue remains the same:

The authors could gather independent, high-resolution outline datasets (often mapped from 1-m or even 0.5-m LiDAR imagery) that do exist in many countries (Switzerland, Scandinavia, Canada, etc), often linked to national surveying efforts. This would provide a largely error-free reference to test GlaViTa independently of GLIMS. The text cited by the authors in their rebuttal from Paul et al. (2013) on errors being independent of resolution relates to satellite imagery and is really not applicable to aerial-based lidar imagery, which is not only much more resolved (sub-meter) but also much more precise (almost no short-scale error correlations, contrary to satellite imagery).

To clarify the potential misunderstanding further: GlaViTa would NOT need to use any of the LiDAR imagery related to these outlines to predict them, but simply use the temporally-closest satellite imagery it is already using and use this outline instead of the GLIMS outline as the ground-truth reference at that time period. Consequently, only outlines would be needed, and many have been digitized throughout the community with authors often happy to share the data (especially if it is just a shapefile and not the lidar imagery, it really is a tiny time investment). A small dataset sample of a dozen glaciers would already be a great independent validation.

This type of validation with very high resolution data such as LiDAR is exactly what is done in other types of remote sensing

estimations (e.g., glacier mass change studies), and would be as beneficial here as it is in these other studies to validate the accuracy of the delineation estimates, including the average IoU (or better, the distance deviation metrics), as well as the range of predicted uncertainties.

3/ Minor comment: Adjusting the title

I don't think the title currently does any favour to the study: "Towards" somewhat implies a next study or a step forward which makes the nature of the findings in this study immediately unclear. A first reader wants to know directly what is the main advance of this study specifically. Additionally, this is a case study (even though spread out through different regions), not a global study. For clarity, the "global" could be kept for a later study.

For instance, the following title seems more appropriate:

"Scalable deep learning glacier mapping using open Earth Observation data matches the accuracy of manual delineation"

(Remarks on code availability)

The repository has improved since the last round of review, with what seems like sufficient details in the README to help other users use the software. I have not run the software.

Version 2:

Reviewer comments:

Reviewer #1

(Remarks to the Author)

The authors have addressed all my remaining comments, and I feel the manuscript is ready for publication.

My only final comments are regarding the GitHub repository, which I have written separately in the code section of the review.

(Remarks on code availability)

While the repository has greatly improved, I still feel it could benefit from a bit more structuring in order to make it more user-friendly and increase the chances of being reused by other researchers. I would suggest the folders and files to be better structured. I understand that the authors have chosen to show folders based on modules. I still think it's best to have a unique folder per package, which inside groups all the folders of the modules. I will not insist on this since it can be quite personal. Nonetheless, I think it would still be a good idea to group all notebooks showcasing the code in a single folder at root level. This can be called `notebooks` for example. Finally, I would also make sure to directly reference the demo Jupyter notebook quite high in the README, so new users can find it easily and get a good overview of how the code works.

Reviewer #3

(Remarks to the Author)

The authors have addressed my previous comments in their entirety and satisfactorily, in particular by performing a new comparison to independent, high-resolution glacier outlines to further back up their study's main claim of being "on par" with manual delineation.

This new comparison is rigorous and its findings are in-line with the authors' previous conclusions, reaching even better accuracy reached than on the other "independent dataset", comparable to that of manual delineation. Additionally, the authors have added the right level of material to both the main text and supplementary to describe this additional analysis, and adjusted the introduction and title to better reflect the study's impact and content.

Thus, in my opinion, this exciting study is ready for publication!

(Remarks on code availability)

The repository is still of good quality, and even further improved.

Response Letter—Towards Global Glacier Mapping with Deep Learning and Open Earth Observation Data

Dear Reviewers,

Thank you very much for the comments and suggestions on our manuscript. We have addressed all concerns raised in the revised manuscript and the point-to-point response provided below.

More specifically, the major improvements are as follows:

- We extended the introduction and the discussion sections, elaborating the broader context and motivation for a wider audience as well as the scientific interpretation of the results
- We enhanced the validation of our model with indirect comparison against human expert uncertainty in terms of area and distance deviations, showing that GlaViTU is on par in the majority of the cases
- We included two case studies demonstrating the applicability of our method for decadal glacier area change analysis
- We moved a technical part of comparing GlaViTU with DeepLabv3+/ResNeSt-101 to the Supplementary, thus improving the readability of the manuscript and making the main message more focused
- We substantially optimised image sizes to improve the reading experience

Response to Reviewer 1:

Reviewer 1: Maslov et al. present a new approach to automatically map glacier extents from multi-source remote sensing data, based on a convolutional-transformer deep learning model. The authors tackle an important problem present in the glaciological community, particularly aiming at improving the temporal resolution and homogeneity of glacier inventories, which would be an important milestone to improve the validation of regional-to-global glacier models. They do so by designing a custom architecture, and by thoroughly assessing different training strategies. This technical complexity represents an important improvement with respect to previous more simple and naive efforts, based on readily-available architectures. The scale of the problem also represents a meaningful improvement compared to the current literature, with previous studies only focusing on reduced regions. For this, this study has a great potential, particularly as an open-source tool which can be progressively refined as a community effort, as suggested by the authors. The fact that the authors share their code in a reproducible manner in GitHub, with a properly structured repository, is a great plus. Moreover, once they share the training dataset, if done correctly, this can serve as a good benchmark for community efforts to try to build on top of this work. This represents the smartest and fastest way to tackle this problem head on, following the principles of open science.

Nonetheless, I believe this study is missing further depth into the analyses of some of the choices and consequences of the results obtained, as well as the bigger picture and the potential of moving towards a reliable and automated method for extracting glacier outlines globally. The paper sometimes feels rather factual and superficial, and it lacks scientific depth beyond the purely technical aspects of the development and training of the model. I have few comments on the technical aspects of the model, which I believe to be generally very solid and well evaluated. However, I would like to see certain aspects of the paper, particularly in the discussion, further developed in order to clearly show both the real scientific potential and impact that this can have for the glaciological and wider Earth observation community, as well as a better understanding on the choices and reasons behind some of the obtained results.

I will address my general comments in GC points, and then I will move to specific line-by-line comments.

GC1: Lack of perspective and "bigger picture"

Due to the wide audience and nature of this journal, I believe it would be interesting to further develop the potential and impact of this work, first for the glaciological community, and then for the Earth observation community.

Starting from the introduction, I believe the current issue with glacier inventories is clearly explained. Nonetheless, I think it would be interesting to better explain why this matters for scientific efforts specifically. There is a lot to be said about how glacier outlines are the baseline product for almost all regional-to-large scale glaciological studies. First of all, remote sensing researchers deriving many glacier products (e.g. ice surface velocities or geodetic mass balance), rely on these outlines. Any biases in those outlines impact all the products used by the community. But that is just the beginning of the workflow, since then modellers

use those datasets to force their models and make assumptions about physical processes or the past or future evolution of glaciers. Moreover, having multi-temporal consistent outlines could further help constrain the evolution and trajectory of glacier models during calibration for past periods. While these is briefly mentioned in the introduction, I think not enough emphasis is put on the importance of this work, somehow diminishing the impact for the wider community.

Some aspects of this could also go in the discussion part, particularly regarding the impact of this work, or at least the potential of tackling this problem in such a general and flexible manner. Another aspect that is missing, is the impact of this work in the Earth observation community. How does this compare to similar efforts for other Earth observation communities outside glaciology? Do any of the lessons learnt here transfer elsewhere? I would further focus on the wider impact of this work and how this could be helpful or impactful to the wider community.

Answer: We deeply appreciate the reviewer's assessment of our work and the contribution of the comments to the quality of our manuscript.

We deliberately chose to keep the results section factual. In contrast, we extended our introduction and discussion to broaden the interpretation of our results for glaciological and EO communities and to further stress the contribution of our work. We discussed in more detail the importance of consistent glacier outline products for glaciologists in Lines 57-82 as well as the methodological value for a wider Earth Observation community in Lines 608-713.

GC2: Limited scientific interpretation of the results

Another aspect that is lacking in this study is more depth in the interpretation of some of the results. While the technical aspects are solid, and the training strategy comparing many approaches is very thorough, one has the impression that some very interesting results are barely analysed or commented, keeping the focus purely on the technical aspects of the machine learning model training. For instance, the insights on the contribution and importance of thermal and SAR are quite limited, mostly just describing how they impact model performance, without digging too much into the physical reasons. Since the bad performance of thermal data comes quite as a surprise, I would appreciate if the authors could further explain why they think this harms model performance, and following this analysis, how it could be adapted in order to correctly leverage this dataset. Thermal data is widely used to study debris covered glaciers, so this somehow contradictory results should be correctly argued and explained. The same goes with SAR, but in a lesser degree. It would be interesting to have further insight on why SAR backscatter and InSAR coherence do help to map glacier termini and artefacts along coastlines, beyond obvious reasons related to cloud penetration.

Answer: Thank you for this comment. We extended our discussion to address these points. We discussed the impact of adding thermal data in Lines 609-620 and InSAR data in Lines 628-631. We conducted additional experiments to explain why adding thermal data harms the performance, particularly for the debris-covered ice, contrary to our expectations based on current literature. For this, we trained regional models for two clusters of regions–debris-

rich (AWA+CAU+HMA) and mostly clean (ANT+GRL+TRP+SAN+SCA)—with and without thermal data, please see the results below:

Feature set	IoU of different subregions													
	ALP	ANT	AWA	CAU	GRL	HMA	TRP1	TRP2	NZL	SAN1	SAN2	SCA1	SCA2	SVAL
Global														
Optical+DEM	0.844	0.949	0.912	0.862	0.937	0.774	0.817	0.903	0.860	0.874	0.958	0.945	0.836	0.936
Optical+DEM+thermal	—	0.951	0.915	0.846	0.937	0.756	0.805	0.895	—	0.863	0.956	0.946	0.857	—
AWA+CAU+HMA (debris-rich)														
Optical+DEM	0.719	0.693	0.897	0.802	0.829	0.735	0.540	0.820	0.812	0.726	0.905	0.903	0.773	0.556
Optical+DEM+thermal	—	0.610	0.909	0.788	0.737	0.739	0.746	0.799	—	0.637	0.888	0.869	0.681	—
ANT+GRL+TRP+SAN+SCA (mostly clean)														
Optical+DEM	0.761	0.952	0.742	0.803	0.936	0.544	0.851	0.906	0.731	0.889	0.958	0.948	0.763	0.921
Optical+DEM+thermal	—	0.952	0.765	0.825	0.936	0.539	0.544	0.904	—	0.891	0.957	0.947	0.888	—

Similar to our previous findings, thermal data does not offer any improvements in a consistent manner. We related it to the limited spatial resolution of the Landsat thermal bands and variability in debris layer thickness that can insulate ice effectively in some cases. Yet, to confirm that, more studies are needed that would employ, e.g., modelled debris thickness data.

GC3: Naming and explanation of testing strategies

Another aspect that could be improved in the study is the naming and explanation of the testing strategies (i.e. "Multitemporal global-scale generalisation"). Two strategies are presented: (1) a test set based on tiles, and (2) a test set based on different regions and time periods (called "standalone"). First of all, I find the names quite confusing, especially with the "standalone" one. I think the authors should better explain each one of them.

If I understood correctly, strategy #1 is the classic approach of randomly splitting the dataset into train, validation and test. Therefore, the test tiles are picked randomly without respecting any spatiotemporal structures in the data. I would be inclined to call this "Randomized test". Whereas strategy #2 is focusing on smaller regions and isolating specific spatial regions and temporal periods for the test dataset. For this case, I would be inclined to call this "Spatiotemporal test". This better reflects the main differences in both approaches, since both approaches are in fact "standalone", as any out-of-sample test dataset should be. Finally, the large difference in dataset sizes between both approaches makes me wonder how reliable are the results of strategy #2. It would be interesting to provide a metric on the number of glaciers or polygons included in each dataset, in order to better understand the robustness of these results. This is seen a little bit in Fig. S1, where there is a much smaller number of data points for the "standalone" approach.

Answer: Agreed. Initially, we also struggled to find good names for the datasets. We renamed them throughout the manuscript as "the tile-based dataset" and "the independent acquisition test dataset"; these names seem to be the most self-explanatory. The tile-based dataset is structured into non-overlapping near-squared 10x10 km² tiles covering 11 diverse regions globally. The tiles are randomly split into training, validation and testing subsets (thus, respecting the spatial structures). The independent acquisition test dataset is specifically used to test the generalisation of our model over different/new satellite image acquisitions and includes data from the Swiss Alps, Scandinavia, Alaska and Southern Canada. This test dataset enables more challenging temporal and spatial generalisation assessments of the

models trained solely on the tile-based dataset. We also added statistics on the number of glaciers and their area (Lines 825-841) for the latter and extended our evaluation by providing area and distance deviations from the reference data (Lines 377-398, 491-500, Table S5) to get a better understanding about the robustness of the method and its performance.

GC4: Example in the GitHub repository

While the repository is work in progress and it is already quite complete, it is hard to reproduce the results due to the high computational requirements of the model. Neither forking the repository myself and running it in local, or even attempting to host this on Binder yourselves are not feasible.

For that, I would encourage the authors to provide a Jupyter notebook with an example (e.g. with the 10% dataset), which shows the baseline workflow on how the model is loaded, trained, and how some results/plots are obtained. Since notebooks can be browsed directly on GitHub, this could provide an easy-to-share showcase demo for people wanting to see how the model works. The notebook could be directly linked in the README for easy access.

Answer: We finalised the GitHub repository (https://github.com/konstantin-a-maslov/towards_global_glacier_mapping). We added a 10% subset of the data to test training on less powerful machines and the pretrained models. After careful consideration of the suggestion, we still decided to organise it as a set of separate scripts rather than Jupyter notebooks that contain all the steps from A to Z. Different users might be interested in different use-case scenarios and automation of their workflows with the use of these scripts, thus more modular file organisation works better here. This structure also simplifies the process for users to create customised Jupyter notebooks when needed based on their specific requirements.

Specific comments

Abstract: "...difficult-to-classify debris...": this sentence is a bit confusing. You are not trying to classify debris, but debris-covered ice, or glaciers. I would rephrase this.

- Agreed. We rephrased the sentence initially. Later, we decided to change this sentence completely for better flow and to make the knowledge gap we address in the manuscript more clear.

L122: The IoU acronym has never been introduced before. I would briefly explain what it is.

- Please see the changes in Lines 158-159.

L140: Why is DeepLabv3+/ResNeSt-101 chosen as a baseline model for comparison? This should at the very least be explained, as it comes quite out of the blue.

- In the preliminary study (Maslov et al., 2023), we already compared our method with several baselines (SETR, ResU-Net and TransU-Net). Here we extend the experimental analysis by comparing it with DeepLabv3+/ResNeSt-101 as one of the state-of-the-art models in computer vision (e.g., also used as a baseline in Strudel et al., 2021). We also added the band ratio method as a baseline for the independent acquisition test dataset (Lines 356-368, Table 1). To make the main message of the paper more focused, we moved the comparison with DeepLab3v+/ResNeSt-101 to the Supplementary.

L152: Indeed, hardcoding spatial coordinates like this might introduces biases. What were those biases in your case?

- *Hardcoding spatial coordinates can lead to overfitting the model to specific terrain features and imaging conditions present in the training dataset. Sensor-specific biases arise due to differences in spectral response and resolution among imaging sensors. Atmospheric conditions such as cloud cover, aerosol content, and varying illumination conditions also introduce variability that the model might learn as features specific to certain locations (when they are provided jointly). These factors can combine to create a model that, while accurate for the training dataset, may not perform as well on new datasets captured under different conditions. We mention these biases in Lines 931-936.*

L153-156: See GC3. Here I would also point the reader to the place in the Methods where this is explained in detail. Overall, this should be done throughout the paper to avoid vagueness and help the reader find the meaningful details, especially for a rather technical paper like this one.

- *Agreed. We added references to the Methods section throughout the paper.*

L159-162: Could you specify how many input features you actually feed into the model? E.g. for optical, do you feed multiple bands? For SAR, I guess it's both intensity and coherence. What about backscatter? Besides specifying the number of input features, I would also refer to a figure/table which gives more details about that. This made me wonder: did you try combining the four types of sources: optical, DEM, thermal and SAR? Could training with both thermal and SAR at the same time help in some way?

- *For optical data, we use only six bands—blue, green, red, NIR, SWIR-1.6 μ m and SWIR-2.2 μ m—to get consistent information from both Landsat and Sentinel-2. For elevation data, we feed into the model two “bands”—elevation and slope. For SAR, we use both backscatter (σ_0) and interferometric coherence, both contain two “bands” as well—the first one is for the ascending orbital path, and the second one is for the descending orbital path (so, four in total). We summarised these details in Lines 797-813 (originally, we wanted to add a table but found that representing it as a list works better) for better clarity.*

As for training Optical+DEM+Thermal+SAR, we cannot perform a comprehensive experiment as the only region where all features are available is Northern Scandinavia (see Table 2). Reporting these results, while possible, would be of little interest from our perspective due to the limited area with predominantly clean ice.

L196-207: See GC2. These results are interesting and meaningful from a wider perspective and a physical/scientific point of view. I would expand this and go deeper in the interpretation in the discussion section.

- Thank you for this suggestion. We discussed it in more detail in Lines 519-536. Debris-covered ice is difficult to classify because of its spectral similarity to the surrounding rocks, also, the inventories we use for training inherit quite some uncertainty for debris-covered glaciers and shadowed ice. Similarly, ice mélange is similar to clean ice spectrally. On top of that, ice mélange is underrepresented in our datasets. Yet, we do not have an intuitive explanation for the coastline artefacts observed in some predictions. We related it to the low reflectance and zero slope, but it remains an open question for us.

Fig. 1: I see 2 losses in the architecture, but I don't recall having read an explanation about the role of each. Where is this explained? This should have its place in the Methods.

- Thanks for pointing this out. Indeed, this point was missed in the original version of the manuscript. Here we use deep supervision where we optimise two identical losses simultaneously, the corresponding explanations were added in Lines 877-878.

Fig. 2: By looking at these tiles I cannot help see that many of them give a very partial view of glaciers, with only a small fraction of them. Have you tried different sizes of tiles for the training? Did you try different strategies to try to maximize glacier coverage within each tile? Have you seen if having a complete view of the glacier in a tile helps with the mapping? Since you are using attention, this could very well be the case. Could you please comment on these aspects?

- We only explored the patch size influence in very early preliminary studies, where we found that 384x384 gives good results. Unfortunately, we do not have enough computational resources to explore the optimal patch size for such a big dataset as presented in the manuscript in a reasonable time. As you mention, we can benefit from more complete views of glaciers due to self-attention, thus, larger patch sizes are likely preferable. We almost hit the upper limit of our GPU memory with 384x384, so we assume it is close to optimal in our setting.

As for maximising glacier coverage within patches, we do not do it as it would require knowing where glaciers are beforehand, and while addressing the problem of mapping them we assume their locations are completely unknown. Though one could design a study where, e.g., RGI outlines are used to produce bounding boxes to find the optimal view, we left it out of the scope of our study.

L244: In the text you mention "tiles" directly. For readers not familiar with computer vision and CNNs, I would introduce this concept.

- Agreed. We introduced tiles and patches in more detail (Lines 185-188).

L248: See GC3. At this point it was quite confusing to follow along the explanations, since it is not very clear what each strategy is and how they differ. I would make sure that this is made clear before carrying on with explaining the results.

- Thanks for pointing this out. We adjusted the explanations and added a link to the methods subsection where it is described in detail (Lines 180-239).

Fig. 3: For this sort of figures, I would explain more in detail what the figure tries to show in the legend. The same goes for Fig. 1.

- Agreed. We enhanced the captions for better clarity.

L301-304: See GC2. Here you correctly point out the outcome of the results, but again, I would go deeper into the analysis in the discussion section.

- Thanks. We extended our discussion according to the comment (Lines 600-623).

L310-322: Same as the previous one.

- Agreed. Please see the changes in Lines 628-631.

L329 "Monte-Carlo dropout" and "plain softmax" need references, and at least some brief introduction. I would also point the reader to the place in the Methods where this is explained in detail.

- We added references for Monte-Carlo dropout and softmax and a link to the methods subsection (Lines 236, 237, 239, 986, 993).

L388: Can be or could be? Are these perspectives or actual possibilities in the current framework that is being presented. This should be made clear.

- These are perspectives we left for further research. We changed the sentence accordingly (Line 537-548).

L391: Same as the previous one.

- Please see corrected in Line 543.

L451-458: See GC2. These are really interesting results. Please go deeper and interpret them in the discussion.

- Agreed. We discussed these results in more depth (Lines 606-623).

L529-536: I fully endorse this community-based perspective built on open-source software. You provide the baseline model to build upon with the help of the community.

- Thank you! We also believe this is the way to foster further advancements in the field.

Fig.: 4 This figure has a lot of potential but in my opinion it is currently underutilized. I would add the percentage of glacier coverage for each region, both in terms of number of glaciers and surface area, and I would also include the total (global) number of glacier coverage (9% according to you), somewhere in the figure. Moreover, it would be interesting to add some sort of pie chart indicating the distribution of clean ice, debris-covered and marine-terminating glaciers. And finally, including the performance of your model(s) - perhaps only the best one - for each region could also provide an overview of how it changes with respect to the type of glaciers. Again, I would make proper use of the legend to explain the figure. It should be standalone, so a reader who skims through the paper can have an idea of what is going on.

- Agreed, we enhanced the figure according to the suggestions. We took debris coverage estimates from Herreid and Pellicciotti.

L580-582: Did you balance the appearance of clean ice, debris-covered and ice mélange in each one of these datasets?

- We made a completely random split expecting that the appearance of clean ice, debris and ice mélange would be balanced between the subsets due to their size (~2700 tiles in total). Implementing a more complicated procedure by intentionally balancing their appearance would require knowing where debris-covered ice and ice mélange are. Unfortunately, we do not have the information about ice mélange directly in GLIMS or RGI.

L608-610: When performing tests for unseen years, were the input features (e.g. DEM, optical) from that new date as well? or only part of them (e.g. optical)?

- All input features were chosen to have the smallest temporal baseline to the inventories of the unseen years/regions, with optical data acquired exactly on the same date. They were completely different from the tile-based dataset. We adjusted the wording a bit (Lines 829-849) to make this point clear.

L662-664: Later on you explain how you use these cycles (e.g. to fine-tune global models). Please explain the general purpose of these cycles here, without going into the details you explain later on in the Strategies section.

- The idea of cycles here is to “shake” the model weights after some time by setting the learning rate to the original value, thus helping the weights to “jump out” from a local minimum or a plateau. For the details, see Loshchilov and Hutter. In this way, a better performance can be achieved in the end as well as a better generalisation as we have several cycles like this.

L665-667: Do you mean after each one of these cycles? It is unclear why do you use them.

- *We evaluate the model after each epoch, not cycle, accordingly the early stopping is applied on the epoch level as explained in Lines 887-888. We agree that the motivation for using these cycles was a bit unclear. We clarified this moment in Lines 880-883.*

L675: While it is good to have all these details here, I found it was hard to understand the nuances of each approach in the main text. I'd rephrase how this is explained there in order to make it clear for the reader, and I would also clearly point the reader to where each thing is explained more in detail in the Methods section.

- *We added the references to the Methods section in Line 225. We have made minor adjustments to the original text (Lines 211-226). We hope that with these references, the connection between the sections is clearer and aids in the understanding of our approaches.*

L713-714: Please provide a reference.

- *Here we cannot provide a reference as this is one of the novelties of our study.*

L745-746: I guess that the parameters of this FFNN were optimized in conjunction with the other parameters of the architecture?

- *Yes, indeed. We clarified this point in Lines 973-974 now.*

L760: I would make a direct reference to this section in the main text to guide the reader.

- *Agreed. We added the corresponding links prior to the Methods section (Lines 204, 208, 225 and 239).*

L767: I would provide a little bit more context on what this is.

- *We added a reference to a DL textbook explaining softmax and slightly adjusted the wording in Lines 993-995 to make it clearer.*

Fig. S1: Why are there so few "standalone"/spatiotemporal data points? It would be interesting to provide a number of independent test and standalone data points, to understand the robustness of the metrics. I guess each point is a region? Since regions are probably larger than tiles, I am not sure if this direct comparison is meaningful.

- *We measure the performance (RMSE and R^2) of the metrics jointly for 556 tiles (from the tile-based dataset) and 4 regions (from the independent acquisition test dataset, equivalent to ~396 tiles of 10x10 km² in total). Here, we did not have an aim to compare them against each other. It is also reasonable from the way Equation S5 is constructed as it does not have any inherent assumptions about the area value/number of glaciers. Excluding the regions does not change RMSE and R^2 as the tiles dominate the population. Calculating things separately for the regions does not make sense to us due to the very limited population size. Here, we present them together on one scatter plot for demonstration purposes.*

We provided the number of points for the test tiles (556) and the generalisation test regions (4) in the caption of Figure S4.

Fig. S5: Something I've realised from your predicted results is that sometimes there can be holes or gaps in a glacier tongue (e.g. Fig. S5e). Such artefacts are mostly 100% unrealistic, since due to glacier ice flow dynamics, it is practically impossible to have such gaps, especially in areas with a continuous ice flow. For that, an automatic post-processing to eliminate such artefacts could potentially improve model performance with a simple solution.

- We agree with this point and appreciate the reviewer pointing this out. We also have it in the discussion: "Post-processing could be applied to eliminate some of the problems. For instance, sliver polygons on the medial moraines of glaciers could be filtered out based on shape descriptors, and artefacts at coastlines could be removed using masks that indicate glacier absence based on a priori knowledge." We are currently investigating these types of misclassification and potential solutions, which will further increase the accuracy of our method in follow-up studies.

Fig. S11: This figure serves to show the spatial testing strategy, but there is no information about time. I would indicate how the cross-testing has been implemented for each one of these tiles, where the model is trained on all but one, and then evaluated in test on a single tile based on an unseen region. Moreover, giving dates or a timeline of how these different scenes at different time snapshots are used for temporal cross-testing would be useful.

- We enhanced the figure and the caption according to the suggestions. To clarify, we never use these data for training the models. The models are trained on the training tiles from the main (tile-based) dataset and then applied to these regions. We better explained this moment in Lines 845-848. With this approach, we mimic the real application case where any available data from a region can be chosen to produce an inventory for a specific year. This input data might vary from the data used in training for this region. This is an advantage as compared to other studies on regional scales, where both training and validation of models are restricted to the satellite data available close to the date of the inventory.

References:

- Maslov, K. A., Persello, C., Schellenberger, T. & Stein, A. GLAVITU: A Hybrid CNN-Transformer for Multi-Regional Glacier Mapping from Multi-Source Data in IGARSS 2023 - 2023 IEEE International Geoscience and Remote Sensing Symposium (2023), 1233–1236.*
- Strudel, R., Garcia, R., Laptev, I. & Schmid, C. Segmenter: Transformer for Semantic Segmentation in Proceedings of the IEEE International Conference on Computer Vision (Institute of Electrical and Electronics Engineers Inc., May 2021), 7242–7252. isbn: 9781665428125. <https://arxiv.org/abs/2105.05633v3>*
- Herreid, S. & Pellicciotti, F. The state of rock debris covering Earth's glaciers. Nature Geoscience 13, 621–627. issn: 1752-0908. <https://www.nature.com/articles/s41561-020-0615-0> (2020).*

Response to Reviewer 2

Reviewer 2: Automated glaciers classification is a challenging task but also an important and valuable one as glaciers integrate climate over decadal timescales and are therefore useful indicators of trends rather than variability.

This is an impressive study in terms of its comprehensive nature and extensive analysis covering a wide range of glaciated regions, model design and tuning, and image types/sensors. It is certainly worthy of publication and I have no substantive comments regarding the methods or findings.

However, as presented, I do not believe it is appropriate for Nat Comms which is a broad readership journal covering all disciplines. In its current form it is more appropriate for an IEEE ML or remote sensing journal such as TGARRS, TIP or ISPRS. The m/s is replete with ML related jargon, often not explained. Terms such as Monte Carol Dropout, SoftMax and IoU (not spelled out) are common to the ML community but far less so to the wider science community. It is not just the terminology used though. The heavy emphasis on comparison to existing ML approaches and performance metrics is common practice in IEEE publications but less appropriate to Nature family journals. There is frequent reference to supplementary figures and material because these are relevant to these performance criteria. The writing style and focus of the m/s is far more in keeping with such more technical journals and it would fit well there and would be a valuable contribution.

Answer: We first would like to thank you for the efforts and comments on our manuscript, which helped us to improve it. We are pleased to read that you generally provide very positive feedback on the manuscript. We also acknowledge your concern about suitability to Nature Communications.

Nevertheless, before submitting the manuscript to Nature Communications, we inquired about the suitability ourselves. After reviewing the scope of the journal, we were confident that the manuscript matches the criteria, as "Nature Communications acknowledges the value of applied science and is also dedicated to publishing high-quality research that represents important technical and technological advances of significance to communities of researchers and professionals working in all fields of applied sciences and engineering" (<https://www.nature.com/ncomms/submit/applied-science-research>).

We also found several publications with a similar scope of introducing advanced deep learning models for various applications, e.g.

- Wu, Z., Zhang, C., Gu, X. et al. Deep learning enables satellite-based monitoring of large populations of terrestrial mammals across heterogeneous landscape. Nat Commun 14, 3072 (2023). <https://doi.org/10.1038/s41467-023-38901-y>

- Mousavi, S.M., Ellsworth, W.L., Zhu, W. et al. Earthquake transformer—an attentive deep-learning model for simultaneous earthquake detection and phase picking. Nat Commun 11, 3952 (2020). <https://doi.org/10.1038/s41467-020-17591-w>

- Andersson, T.R., Hosking, J.S., Pérez-Ortiz, M. et al. Seasonal Arctic sea ice forecasting with probabilistic deep learning. Nat Commun 12, 5124 (2021). <https://doi.org/10.1038/s41467-021-25257-4>

- Wang, Z., Majumdar, A. & Rajagopal, R. Geospatial mapping of distribution grid with machine learning and publicly-accessible multi-modal data. *Nat Commun* 14, 5006 (2023). <https://doi.org/10.1038/s41467-023-39647-3>

- Yeh, C., Perez, A., Driscoll, A. et al. Using publicly available satellite imagery and deep learning to understand economic well-being in Africa. *Nat Commun* 11, 2583 (2020). <https://doi.org/10.1038/s41467-020-16185-w>

...

Our work presents a significant step forward in the methodology of fully automated glacier mapping on a large scale, and our results approach human expert quality, as shown in the manuscript by comparing with existing glacier inventories. This advancement is crucial for the cryospheric and climate science communities, as articulated in the manuscript. Hence, we concluded that our manuscript fits the general scope of Nature Communications.

In the new version, we improved the readability for a wider audience, highlighting the broader implications of our work and moving some very technical parts to the Supplementary completely. We also acknowledge your concern about jargon and terms used in the machine learning community and did our best to explain the terms and improve readability and understandability to a general readership. This is a multi-disciplinary study which involves and addresses experts from machine learning, glaciology, remote sensing and data science. In such a case, there will always remain a dilemma between being scientifically precise and using field-specific terminology (with terms even having different meanings in different research fields), while maintaining good readability for an even broader audience.

Reviewer 2: The paper goes from introduction straight to results bypassing data sets altogether, which does not appear until the methods. Crucially there is no discussion anywhere in the main text what the source of the ground truth was and this appears to be manually derived inventories from different sources acquired at different dates. It is unclear if the authors matched the imagery used to the dates of the inventories and with what latitude. This is important because some of those inventories likely cover more than one year. How long a time gaps is acceptable?

Answer: In writing the manuscript, we followed the general template of the journal. Yet we understand the concern of the reviewer and added a short description of the datasets before the results, please see the corresponding changes in Lines 180-204.

Reviewer 2: Related to this is the complete lack of any discussion of the timing of the imagery which is crucial for reliable classification. Late summer is needed to reduce the effects of variable seasonal snow cover that is difficult to impossible to differentiate from glacier if present. This issue is not discussed at all, which I found surprising. Despite no mention of this, the results are excellent but it is unclear if this is because of careful choice of image data/season or not?

Answer: The reviewer is correct that this is not explicitly mentioned and discussed in the manuscript. In the Method section, we explain loosely: "The data sources were chosen based upon the data availability and the date for which inventories were created." What seemed very natural for us as the basis for a well-done study should have been explained more

precisely. We elaborated on the fact that we chose the satellite data as close as possible to the date on which the inventories were made, always as close as possible to the end of the ablation season before the first major snow event to ensure minimal seasonal snow cover (see Lines 776-784). For optical data, we chose an image of the exact data as in the inventory metadata if available. If a publication describing the optical data preparation procedures was available (New Zealand case only), we were following exactly the same routine. If metadata did not specify either the source data or a publication, we did not use these data as a reference. We also discarded the tiles for which the glaciers were mapped based on several optical images acquired at different dates to avoid time-consuming manual treatment of these areas. For SAR data, we have a higher temporal baseline of less than a week for Sentinel-1 (to less than a month for ENVISAT due to the lower temporal resolution). For digital elevation model –up to a maximum of seven years (except the polar regions, where SRTM is not available, see Lines 783-784). On the one hand, we believe that this affects the accuracy of the method negatively, but since glaciers in general do not change significantly within weeks or months this decrease is likely restricted to the termini of the glacier where most changes happen. On the other hand, we believe that this adds some noise to the data having a regularising effect, thus potentially preventing overfitting and aiding the generalisation of the model, overall contributing positively to the final application on “unseen data.” Yet, this effect cannot be quantified.

Response to Reviewer 3

Reviewer 3: The authors present a very interesting study utilizing deep learning for glacier outline delineation, tested under varied calibration schemes, imagery types, spatial and temporal scales, and glacier surface conditions worldwide. The paper focuses largely on methodological development, improvement and optimization of the methodology, and the release of a related dataset for 9% of glaciers.

While the study is presenting very valuable work and is well-written, I am not sure if it fits for this type of journal, for the simple reason that it is very methods-focused and thus hard to read/access for a wider audience not specialized in machine-learning. It might be a better fit for a methods journal.

Answer: Thank you for your valuable comments and encouraging feedback on our work, they have been essential in enhancing the manuscript.

We also acknowledge your concern about the suitability for Nature Communications. Before submitting, we carefully considered the journal scope and are confident that our manuscript meets its criteria for publishing impactful research in applied sciences and engineering, as outlined in their submission guidelines: "Nature Communications acknowledges the value of applied science and is also dedicated to publishing high-quality research that represents important technical and technological advances of significance to communities of researchers and professionals working in all fields of applied sciences and engineering" (<https://www.nature.com/ncomms/submit/applied-science-research>).

We also found several publications with a similar scope of introducing advanced deep learning models for various applications, e.g.

- Wu, Z., Zhang, C., Gu, X. et al. Deep learning enables satellite-based monitoring of large populations of terrestrial mammals across heterogeneous landscape. Nat Commun 14, 3072 (2023). <https://doi.org/10.1038/s41467-023-38901-y>

- Mousavi, S.M., Ellsworth, W.L., Zhu, W. et al. Earthquake transformer—an attentive deep-learning model for simultaneous earthquake detection and phase picking. Nat Commun 11, 3952 (2020). <https://doi.org/10.1038/s41467-020-17591-w>

- Andersson, T.R., Hosking, J.S., Pérez-Ortiz, M. et al. Seasonal Arctic sea ice forecasting with probabilistic deep learning. Nat Commun 12, 5124 (2021). <https://doi.org/10.1038/s41467-021-25257-4>

- Wang, Z., Majumdar, A. & Rajagopal, R. Geospatial mapping of distribution grid with machine learning and publicly-accessible multi-modal data. Nat Commun 14, 5006 (2023). <https://doi.org/10.1038/s41467-023-39647-3>

- Yeh, C., Perez, A., Driscoll, A. et al. Using publicly available satellite imagery and deep learning to understand economic well-being in Africa. Nat Commun 11, 2583 (2020). <https://doi.org/10.1038/s41467-020-16185-w>

...

Our work presents a significant step forward in the methodology of fully automated glacier mapping on a large scale, and our results approach human expert quality, as shown in the manuscript by comparing with existing glacier inventories. This advancement is crucial for

the cryospheric and climate science communities as articulated in the manuscript. Hence, we concluded that our manuscript fits the general scope of Nature Communications and hope you will reevaluate your initial assessment of suitability.

Reviewer 3: In terms of impact, I think that either an application extension using this method for all glaciers globally, or a methods comparison evaluating against traditional mapping methods in detail would allow to draw clearer conclusions for a larger community.

Answer: We fully agree that the impact of the paper would be even higher with a global application. This is planned for the future, after addressing the shortcomings, hence the title “Towards global glacier mapping...” We still believe that the methodological advancements are worth publishing for a general audience beyond glaciology and machine learning now and in Nature Communications. We anticipate high scientific impact as we provide many advancements compared to the state of the art. Improving the datasets and methods, as well as the automation will take months, but the GlaViTU model and the datasets can already be useful for the cryospheric community and beyond, speeding up the community effort for high-quality datasets.

Reviewer 3: In this study, the developments are valuable but main conclusions somewhat limited:

- In terms of glaciological knowledge, we do not yet learn anything from newly derived datasets (limited to 9% of glaciers, and the authors do not analyse the area changes that can be estimated from the multi-temporal outlines, they are probably keeping this for later scaling efforts),

Answer: This is true. As mentioned above, we are working on implementing it at a large scale. This study represents two years of work and for the original manuscript, we decided against this approach as it blends the current and upcoming study. It is also true that there is a lack of advancement of glaciological knowledge, this was not the purpose, the advancements are of technical sorts, similar to the methodological studies on other topics published in Nature Communications mentioned above.

Yet, we claim that the methods we propose here can already be used for temporal analysis of area changes as is. We conducted two case studies in the Swiss Alps (Aletsch complex, 1998 vs 2023) and in Southern Norway (Hardangerjøkulen, 1996 vs 2022) to demonstrate that. The Aletsch complex diminished in area from 340.01 km² (253.12–439.83 km² taking into account the confidence bands) to 257.48 km² (179.53–350.43 km²) for the indicated period. Similarly, Hardangerjøkulen shrank from 77.01 km² (69.65–90.09 km²) to 64.48 km² (58.05–71.88 km²). Also, see the closeups below:

Top panel: glacier retreat of the Aletsch complex. Bottom panel: same for Hardangerjøkulen. Orange indicate glacier outlines and 95% confidence bands for the later acquisition date, and pink—for the earlier. The basemap images are from the later dates. Note wider confidence bands for debris-covered and shadowed ice.

These case studies demonstrate the capacity of the model to capture significant glacier area changes. We included these cases as a separate section in the supplementary (S.1) and briefly discussed them (Lines 590-596). In future work, we plan to perform area change analysis first for the whole of Europe and then globally (after some methodological improvements, possibly from the community as well, and with more advanced datasets, e.g., including InSAR). This, however, is out of the scope of this particular paper.

Reviewer 3: In terms of methodological development for glacier delineation, while the methods are showing very promising results for large-scale automation and we learn a lot from comparing different calibration/data/scales, the authors do not reach substantial conclusions on the improvement compared to current semi-automated methods and published datasets.

Answer: Comparing to traditional methods from a quality perspective was not the aim of the study as they are not capable of the final goal of fully-automated multi-temporal global glacier mapping with high efficiency and low cost. Instead, we used the results from traditional semi-automated methods, i.e. GLIMS and RGI entries, as quality measures to compare to and found very good accuracy for clean-ice glaciers and good, yet improvable, in several other cases, e.g. for debris-covered glaciers.

To clarify this, we wrote in the introduction “Simple thresholds of optical band ratios are effective methods to map clean-ice glaciers on different scales (14-16). The choice of threshold values, however, can vary for different imaging conditions (17) and within a scene (14). Moreover, threshold-based methods fail to classify complex glacier parts such as debris- or vegetation-covered ice requiring labour-consuming manual corrections or application of more sophisticated methods (18).”

To get a better understanding of how much correction is needed after applying widely used band ratio methods and our models, we added a comparison in terms of IoU in Table 1.

Please note that our method also yields confidence estimates that can be used to point out the problematic places and thus guide manual corrections, while the band ratio method does not. So, our models can become a valuable tool for remote sensing specialists in glaciology.

We also tried to compare the uncertainties of our results with the human expert uncertainties reported in Paul et al. (2013) and GLACE experiments (GLIMS book, Chapter 7).

*Unfortunately, the vector files were not provided by the authors to make a direct comparison. However, we evaluated our model outputs in terms of distance deviation and areal deviation from the reference data (output of **one** human expert) for 17 representative glaciers of different sizes and debris coverage. Please see these results in Table S5. Our results suggest that we are if not on par with human experts but at least approaching their uncertainty quite closely. Though here we need to highlight some points:*

- In GLACE, they also estimate the uncertainty of ice divides, where the uncertainties are usually higher in their case, not only ice complex boundaries;*
- All these studies report results for very limited populations of glaciers, potentially not representative of the whole of GLIMS and not as exhaustive for the experts;*
- None of them include large debris-covered glaciers or large shadowed areas, which present a large source of errors in inventories;*
- In such a setting, where the experts know that their results will be compared against each other, they are more motivated to “do their best”.*

Hence (except for the first point), one should expect overoptimistic estimates of human expert uncertainty from these studies, and the actual manual mapping uncertainty is not well investigated on large scales—a huge knowledge gap. Indeed, in our case, we observe human expert uncertainty of several kilometres in different scenarios (e.g., ~2 km deviations in Figure 2 i or ~3 km in Figure S10 d as revealed by looking at high-resolution images), which is ~5–10 times more than reported by Paul et al. (2013) for debris-covered ice. We discussed these points in Lines 487-518. Please find some examples from Paul et al. (2013) of human expert inconsistencies while manually mapping glaciers from high-resolution and Landsat images below:

[REDACTED]

Multiple digitisations of glaciers by different experts taken from Paul et al. (2013).

[REDACTED]

Our results for the same glacier in the Alps derived from a Landsat 5 image as presented in Paul et al. (2013).

In short, the study currently does not help answering any of the two following main questions it relates to:

- (Glaciological focus) How much has glacier area changed in the past decades (regional changes, etc)?

Answer: Yes, this is true and, as explained in detail above, it is not the purpose of the study. Our work proposes significant methodological advances, see explanation below. Yet, the methodological advances proposed in this paper are fundamental for enabling future studies to pursue multitemporal glacier area change analysis, as illustrated in the example provided above.

- (Methodological focus) Can we map glaciers with deep learning better, or with equal confidence, than with previously semi-automated (=manually-corrected) delineation methods and existing datasets?

Answer: Given the discussion above and the new validation results provided, we claim that our model is at least approaching human expert uncertainty, often being on par with it.

Perhaps, a more important question is which method is the “best” for global multitemporal glacier mapping, where “best” does not only mean the outline quality but also cost, expenditure of human labour and consistency.

Since it might take months to produce a region-scale inventory and years for a new global inventory from satellite imagery in a manual or semi-automated manner at a high financial cost and expenditure of human labour as well as large international coordination, e.g. through RGI, we are certain that the next global glacier inventory will be based on deep learning. Also, while the quality argument is valid for single outlines or regional inventories, the ambiguity in image interpretation often leads to errors and systematic biases, restricting the utility of manual or semi-automated inventories for multitemporal analysis when derived by different experts. Our approach, unlike human experts, yields fully reproducible results, thus it will be consistent across regions and time periods—a must for change analysis studies. Hence, this study represents a huge methodological advance as it presents a deep learning model which is for the first time capable of multi-temporal glacier mapping at a quality approaching human experts, low cost and low expenditure of human labour (which in return will be available for other important research) with high consistency and advanced accuracy assessment.

My main criticism is of course with the second question (the first one being a choice of the authors to not scale yet and focus on methods), for which I have some detailed comments below.

Main comment 1: Reference data to evaluate accuracy and precision more reliably?

The authors point out in their main text (and more extensively in their supplementary materials) that there is an absence of reference data, which they circumvent in the rest of the study by implementing a confidence-based IoU estimation. The approach is interesting but, from that point on, it is somewhat hard to assess how much the confidence values can really be trusted, and most importantly how the results might compare to semi-automated mapping (which could be fairly easy to run against the deep learning approach).

Answer: We believe this is not fully understood by the reviewer, and we like to better clarify it. In fact, there is a large dataset available for training, validation and testing of the model (GLIMS/RGI, described in the Methods section). All the IoU measures reported in the paper are derived by comparing our results with these reference data produced by human experts.

*Yet, if a future user wants to apply the model to a specific glacier at a specific time, there is a high likelihood that no reference data is available for accuracy assessment (e.g., 2024 inventories immediately after the satellite data becomes available). Therefore we implemented the confidence-based IoU estimation to enable the user to get a valuable accuracy estimate. These confidence-based IoU estimates are **only** reported in Figure S4 and compared with the actual IoU values. We better clarified this point in Lines 1525-1532.*

Furthermore, we thoroughly tested the quality of our confidence estimates against the reference and found that they are nicely calibrated and the interpretability and reliability of results are also increased. We also evaluated the confidence estimates on the independent acquisition test data and obtained similar results (Lines 469-476). These uncertainty estimates are either valuable to support manual corrections of glacier outlines (on a local to

regional scale), as input for glacier models, or as input for more automated postprocessing e.g. of time-series of annual glacier inventories derived with GlaViTU.

Reviewer 3: There are several regions where glacier outlining is done at very high resolution based on local aerial datasets and extensively checked manually (golden example: the Swiss Alps, e.g. GLAMOS and Linsbauer (2021)), which could be used as “reference” to compare to temporally-close estimates from your method, and compare against more classical methods.

*Answer: While this is true, this is also not the purpose of the study, as a globally consistent dataset of aerial imagery for glacier mapping is not in sight (one can be very sceptical if it is possible at all). Datasets with high resolution (e.g. Planet or Maxar are potentially available globally yet at high costs). Furthermore, Paul et al (2013) claim: “Results show a high variability in the interpretation of debris-covered glacier parts, **largely independent of the spatial resolution** (area differences were up to 30%)”. They also cite other studies showing systematic biases between inventories digitised from high- and medium-resolution images limiting the meaningfulness of the comparison further: “The results presented above confirm previous accuracy assessments (e.g. Paul and Kääh, 2005; Bolch and Kamp, 2006; Andreassen and others, 2008; Paul and others, 2011) that reported relative area differences of the automatically derived outlines from manually digitized outlines on higher-resolution datasets between 2% and 5%.”*

Alternatively, the authors could manually delineate/adjust a few dozens glaciers (or even hundreds, as done by some terminus mapping studies over Greenland/Antarctica) across the wide range of surface conditions and acquisition dates that they want to test, to create their own reference dataset, possibly using local datasets of higher resolution to ensure limited manual errors.

A reference dataset, even limited, would be invaluable to be more conclusive when stating how good an improvement the data learning approach makes compared to traditional methods, and what we can expect from scaling.

*Answer: From a quality perspective manual or semi-automated mapping will probably stay the golden standard for automated method validation, even given the inherent uncertainties as shown above. We **use the human-derived inventories (GLIMS/RGI) in large quantities** for our model validation (see above). We even added a secondary dataset to prove the applicability of the model on data from other regions and the same regions but from different sensors and of other regions. Nevertheless, manual or semi-automated mapping methods are not fast or easy to scale to reach multitemporal global glacier mapping.*

*However, as we **only** compare our results with manually derived inventories, we cannot claim any improvement by the model **over** them in terms of accuracy. Yet, we can claim other, perhaps even more important, advantages—consistency and full reproducibility. An intersection over Union IoU = 1.0 would mean perfect overlap between the reference outlines and the outlines derived by GlaViTU. In many regions we are significantly above IoU = 0.9, indicating very good performance. An indirect comparison with human expert*

uncertainties was discussed above. Hence, we approach the quality of manual and semi-automated mapping by experts.

Main comment 2: Spatial considerations for IoU metric

It is currently unclear how the IoU metric is aggregated in space. Errors in delineation are typically a function of the instrument resolution, but not the glacier size, so they shrink with increasing glacier size when measured in percentage (what the IoU does per-glacier).

The authors report a much better performance in regions with large glaciers (Antarctica, Svalbard, Andes, Svalbard), and a lesser performance for regions with smaller glaciers, which lets me think that maybe the IoU was not area-weighted per-glacier when reporting regional-scale values?

In any case, a dependency on glacier area likely exists and should be accounted in the analysis of the results, as well as other variables of interest (%age of debris cover?) to discuss the result.

Standardizing depending on the glacier area would also allow metrics to be comparable between different regions.

Answer: The reviewer makes a good point here. The main difference in performance likely stems from the surface type of the glacier, with debris cover the most challenging to distinguish from surrounding rocks. It might be partially related to the size as larger glaciers will tend to have larger areas of clean snow and ice.

We conducted an experiment (see the table below) using synthetic outlines where we modelled delineation errors of different scales for a subset of glaciers with varying sizes. We did so by creating buffers of sine waves (both positive and negative) along the boundaries with different amplitudes, please see an example:

Dorothy glacier outlines. Left: original. Right: after adding +/-100 m sine-wave buffer.

Indeed, there is a relation between IoU and the reference polygon areas. Unfortunately, this relation will also depend on the glacier compactness and shape in general, which are difficult to quantify jointly.

glacier	Area, km ²	+ -10m, IoU	+ -20m, IoU	+ -30m, IoU	+ -40m, IoU	+ -50m, IoU	+ -60m, IoU	+ -70m, IoU	+ -80m, IoU	+ -90m, IoU	+ -100m, IoU
Brattbreen	0.152900	0.899121	0.787449	0.683499	0.588216	0.500442	0.413279	0.331762	0.260672	0.201603	0.149349
Gamchigletscher	1.458283	0.960259	0.917830	0.879564	0.843579	0.808962	0.775094	0.741651	0.708430	0.675659	0.643397
Tundraskarsbreen	3.231000	0.973853	0.945544	0.919660	0.895033	0.871575	0.848537	0.825111	0.801772	0.778505	0.754980
Langgletscher	9.114895	0.973335	0.944871	0.919522	0.896022	0.873712	0.852449	0.832190	0.812945	0.794497	0.776927
Dorothy glacier	9.185473	0.969287	0.936614	0.907016	0.879110	0.852378	0.826775	0.802541	0.779590	0.757695	0.736440
Oberaletschgletscher	18.938481	0.969873	0.937299	0.908189	0.881406	0.856424	0.832830	0.810178	0.788373	0.767327	0.746884
Kilippi glacier	22.396888	0.983855	0.966454	0.950423	0.935185	0.920530	0.906371	0.892543	0.878963	0.865597	0.852451
Unteraargletscher	24.632215	0.975920	0.949867	0.926515	0.905202	0.885254	0.866404	0.848428	0.831191	0.814645	0.798681
Tonsina glacier	40.688209	0.986394	0.971031	0.956828	0.943414	0.930654	0.918404	0.906641	0.895357	0.884517	0.874108
Scimitar glacier	41.038268	0.976774	0.951806	0.928923	0.907234	0.886477	0.866410	0.847004	0.828302	0.810368	0.793216
Tunsbergdalsbreen	46.044800	0.989189	0.977466	0.966679	0.956437	0.946608	0.937121	0.927911	0.918947	0.910159	0.901532
Stephens glacier	49.793461	0.985155	0.968526	0.953142	0.938597	0.924749	0.911478	0.898659	0.886204	0.874196	0.862525
Tiedemann glacier	58.890293	0.980384	0.958978	0.939150	0.920325	0.902271	0.884958	0.868361	0.852366	0.836957	0.822097
Aletschgletscher	82.278448	0.986707	0.972365	0.959581	0.947907	0.936944	0.926522	0.916504	0.906858	0.897497	0.888340
Marcus Baker glacier	173.755546	0.988379	0.975342	0.963349	0.952087	0.941412	0.931186	0.921278	0.911633	0.902242	0.893061
Matanuska glacier	319.141694	0.990422	0.979744	0.969937	0.960700	0.951893	0.943430	0.935237	0.927267	0.919491	0.911876
Kliniklini glacier	469.909866	0.990566	0.980272	0.970711	0.961622	0.952928	0.944546	0.936414	0.928495	0.920763	0.913196

IoU estimates for modelled polygons against the reference polygons of different areas with varying artificial delineation ‘error’. “+ - x m” should be read as adding a buffer to the reference polygons with the sine-wave size varying from -x to +x meters along the boundary.

Nevertheless, we find that the size of the glacier does not necessarily play a major role in our results, e.g. large glaciers with substantial debris cover in High Mountain Asia still have a lower IoU, while small glaciers in New Zealand (with high debris percentage) and the tropics (mostly clean) are mapped with higher accuracies. A similar situation is seen in Table S5 (debris coverage explains more variance of the IoU values as compared to area, $R^2 = 0.60$ vs 0.16). Also, as we report the aggregated IoU over a whole region (i.e. for all pixels at once), the effects of the small areas shown in the experiment above are negligible (except maybe TRP1 and SCA2, but those show relatively good performance). Instead of inventing new non-conventional metrics, we additionally investigated glacier size-related performance in more detail and reported the results in terms of detection accuracy, distance deviations and glacier areas (please see Table S4). The only point we found related to size is that small glaciers ($< 0.1 \text{ km}^2$) are unreliably detected by the model. We discussed this fact in Lines 521-528.

To provide an even more intuitive interpretation of IoU values for specialists with no machine learning/statistics background, we show here that IoU sets the upper bound for the relative area difference (δ) as follows:

$$\delta \leq \frac{1}{IoU} - 1$$

*The upper bound of the relative area difference is set by IoU. The modelled values are obtained by generating random sets of **artificial** references and predictions.*

These results are easy to derive starting from $IoU = TP / (TP + FP + FN)$ and $\delta = |(TP + FP) - (TP + FN)| / (TP + FN)$, and treating δ as a function of both FP and FN. Yet, we found these upper bounds are rather pessimistic for our results (e.g., the results from Table S5 are, on average, 53% better compared to the theoretical upper bound) and thus did not include them in the manuscript.

We added estimates of %age of debris cover from Herreid and Pellicciotti (note that these are approximate due to the temporal misalignment and also contain errors, e.g., missing some debris in High Mountain Asia) next to the previously presented result.

Line-by-line comments:

L8: “while” doesn’t fit this sentence, “and”?

- *Agreed. Please see the changes on Line 8.*

L19-23: Careful not to oversell the importance of glaciers for freshwater, it is limited and often exaggerated in introductions, see Gascoin (2023).

- *Thank you for pointing this out. We made the corresponding changes (Lines 20-24).*

L24-26: GLOFs are actually decreasing in magnitude, see for example Veh et al. (2023).

- *We adapted our text according to your suggestions (Lines 24-29) and added the reference, thank you for pointing us to the latest research.*

L122: Define Intersection over Union acronym

- *Please see the changes in Lines 157-159.*

L335/Fig.3: Define Expected Calibration Error acronym.

- The acronym is already defined in Line 437. We added its definition to the caption of Figure 3 as well.

The text is very well written, but I think the authors would benefit from adapting some of the language they use for a broader audience of scientists in cryospheric sciences, or clearly defining technical terms early in the text.

Answer: Agreed. We did our best to avoid the use of acronyms for improved readability and tried to address the remaining issues carefully, please see the revised manuscript. This is a multi-disciplinary study which involves and addresses experts from machine learning, glaciology, remote sensing and data science. In such a case, there will always remain a dilemma between being scientifically precise and using field-specific terminology (with terms even having different meanings in different research fields), while maintaining good readability for an even broader audience.

References:

Linsbauer, A., Huss, M., Hodel, E., Bauder, A., Fischer, M., Weidmann, Y., Bärtschi, H. & Schmassmann, E. 2021, The new Swiss Glacier Inventory SGI2016: From a topographical to a glaciological dataset. *Frontiers in Earth Science*, 22, doi:10.3389/feart.2021.704189.

Gascoïn, S. (2023). A call for an accurate presentation of glaciers as water resources. *WIREs. Water*. <https://doi.org/10.1002/wat2.1705>

Veh, G., Lützow, N., Tamm, J., Luna, L. V., Hugonnet, R., Vogel, K., Geertsema, M., Clague, J. J., & Korup, O. (2023). Less extreme and earlier outbursts of ice-dammed lakes since 1900. *Nature*, 1–7.

References:

*- Paul, F. et al. On the accuracy of glacier outlines derived from remote-sensing data. *Annals of Glaciology* 54, 171–182. issn: 02603055 (July 2013).*

*- Raup, B. H. et al. Quality in the GLIMS Glacier Database. *Global Land Ice Measurements from Space*, 163–182. https://link.springer.com/chapter/10.1007/978-3-540-79818-7_7 (2014)*

*- Herreid, S. & Pellicciotti, F. The state of rock debris covering Earth's glaciers. *Nature Geoscience* 13, 621–627. issn: 1752-0908. <https://www.nature.com/articles/s41561-020-0615-0> (2020).*

Reviewer #3 (Remarks on code availability):

The repository is not fully ready but already in very good state: detailed README, data links, install guidelines, hardware notes, script descriptions, function modules nicely sorted.

Answer: In the meantime, we addressed the remaining issues. Please visit the updated GitHub repository: https://github.com/konstantin-a-maslov/towards_global_glacier_mapping

Response Letter—Scalable Glacier Mapping using Deep Learning and Open Earth Observation Data Matches the Accuracy of Manual Delineation

Dear Reviewers,

Thank you for the comments and suggestions on our manuscript. We have addressed all the concerns raised in the revised manuscript, as detailed in this letter.

The changes as compared to the previous version include:

- We revised the title
- We added a section in the Supplementary comparing our model outputs with the Swiss Glacier Inventory 2016 reference data
- Several figures were adjusted and relocated
- We softened the tone about the scientific impact of our work in the introduction
- Some of the technical details were moved from the result description to the methods section
- We showed that our method matches the accuracy of manual delineation and is accurate enough to resolve decadal glacier changes and, hence, will enable fulfilling the GSOC standards for estimating regional glacier area change for all glacierised regions in the near future

Please find more details below.

Response to Reviewer #1:

The authors have substantially updated the manuscript, following most of my comments as well as other input from the rest of the reviewers. The manuscript has gained in depth, particularly in some scientific analyses that I asked for; while some technicalities and complexity have been removed from the main text in order to better suit the journal format of Nature Communications. Nonetheless, I still feel that the manuscript needs a few modifications in order to improve its readability and to convey the scientific message of the paper in a clearer manner. As I mentioned in my previous review, the technical aspects of the paper are solid, but the message needs to be properly adapted and the story-line streamlined in order to fit in the narrative of this journal.

Answer: We sincerely appreciate the reviewer's assessment and the feedback, which helped us to improve the quality of the manuscript further.

I have listed different comments across the paper, which relate to the above-mentioned topics. Overall, my comments go in the direction of these last changes, in order to improve the scientific message and to remove unnecessary technical details from the main text. Once these changes are addressed, I don't foresee any more changes blocking publication from my side.

L205: It should read "We propose ...".

Answer: Agreed. Please see this change on Line 200.

L369-398: While the addition of this analysis for small glaciers is interesting, it suffers again from too many technical details which dilute the flow and tone of the paper. I would suggest keeping the most important results in there, focusing on the scientific implications, and move all the technical details to the methods or the supplementary material.

Answer: Agreed. We moved the technical details to the methods section (Lines 351-378). There we arranged a separate subsection on accuracy assessment (Lines 1009-1035).

Fig. 1. I see you have added a little bit more explanations in the figure, but this figure remains highly technical for such a wide audience. Only computer vision experts will make sense of this figure, so I suggest two things: (1) Update the figure by adding envelopes or some visual trick to distinguish the role of each part of the network. The goal here would be to "translate" each part of the architecture to the scientific/applied task which can be understandable by a wider audience; (2) Add these extra explanations in the caption as well. This figure should be much more self-explanatory if it has to remain in the main text.

Answer: As the reviewer suggested, we added envelopes to the figure and expanded the caption (see below). Now it clearly clarifies which part of the model is responsible for what. We also moved the figure to the Supplementary as it is indeed a bit too technical for this type

of journal. Instead, we lifted up what was Fig. 4 to have a better overview of the data and the results earlier in the text.

Fig. S10 Glacier-VisionTransformer-U-Net (GlaViTU). Boxes and numbers in them represent tensors and their shapes in the height \times width \times channels format, and arrows indicate operations and data flow. GlaViTU **I** fuses multi-modal inputs such as satellite images and elevation models, **II** extracts global features with a vision transformer, **III** refines local features with a convolutional subnet and **IV** yields the final classification map.

Regarding my previous comment on the partial coverage of tiles for glaciers. I understand that you are hitting the memory limit of GPUs with the largest possible tiles. Nonetheless, could you verify if the performance is better for already existing tiles that happen to cover a full glacier? This could be an interesting hint. You could relate this to the metrics you have shown where your model performance is much better for large glaciers. This is interesting, because for large glaciers you will almost never have a complete view of the glacier in the tile, so from a technical point of view, it could become equivalent to just seeing a smaller glacier. It would be interesting to disentangle this and to understand exactly why you are obtaining these results.

Answer: Thank you for this suggestion. We delved deeper into this question. We took the Swiss Glacier Inventory 2016 data (as we were working on it closely for this revision, see the details in S.2) and selected ice complexes that are small enough to fit one patch size but big enough to guarantee a partial view when the window is shifted by half the patch size. For the latter situation, the model was run four times for all possible partial views, and the predictions were aggregated by averaging as it is done during regular inference.

*In total, we got **n=43** ice complexes that met the size criteria. We checked whether the IoU values between complete and partial view experiments were significantly different from each other with the Wilcoxon test (as we could not assume the normality of the sample, yet it does not change the conclusions if one naively runs a regular t-test for paired samples): $W=429$, **p-value=0.603** (> 0.05). This analysis allows us to state with high confidence that there is no difference between running the model on complete and partial views of glaciers. That would imply that the features the model can extract from the partial views of the glaciers are enough in most cases. For more discussion on how glacier size affects the mapping quality, please also see the previous rebuttal.*

Some qualitative illustrations are provided below:

Comparison of the results derived from complete and partial views of ice complexes. Partial view window sizes are altered slightly for visualisation purposes. The view windows are $3840\text{ m} \times 3840\text{ m}$.

As we did not pose this question in the paper (which is on the long side already) and these results do not present new significant conclusions, we decided to keep them in this response letter only.

Fig. 3. I see you have expanded the legend, but this figure still feels way too obscure. Again, I think you should really try to make Figures as self-explanatory as possible. Try to design them in a way so readers quickly browsing the paper can make sense of that topic even without reading the full text. You mention before and after calibration. Calibration of what? Model parameters? Hyperparameters? Also, explain the results. Don't let readers interpret them by themselves only. What are you trying to convey here? What relationships can we see, and why is this important?

Answer: Agreed. We further expanded the caption, which should make the figure self-explanatory now (see below). The idea behind confidence calibration is to map the values of confidence to the values of expected accuracy so the former can be interpreted in absolute terms. After proper calibration, one can expect that $X\%$ of the pixels predicted with confidence $X\%$ are predicted correctly. Without this step, we can only compare predictions to each other (i.e. this pixel prediction is more confident as compared to that one).

Fig. 3 Reliability diagrams for confidence derived with Monte-Carlo dropout: a before and **b** after predictive confidence calibration. ECE stands for expected calibration error, lower values indicate better calibration. Bins and orange curves depict the actual accuracy versus confidence as evaluated on the validation subset, and green lines show the ideal calibration case. After calibration, confidence aligns more closely with the actual accuracy, which enables interpreting predictive confidence in more absolute terms. In this particular case, one can expect that $X\%$ of the pixels predicted with confidence $X\% > 60\%$ are predicted correctly. Without this step, one can only compare confidence levels of predictions to each other.

GitHub repository: While the repository is quite complete, and the README covers quite a lot details, I still feel that the files are quite spread out and it generally lacks a bit of structure. Instead of dumping so many files at the root level of the repository, I would encourage to further structure this into folders, and to clearly separate the internals from the scripts and API that the user needs to use. This means, for example, to place the scripts for the workflow in a single folder, and to reduce the already existing number of folders by further grouping them into folders. While I understand that for pure processing reasons it is much better to have the files as .py scripts, I still think it would be nice to have a Jupyter notebook already executed, just for showcasing purposes. I have browsed and opened the scripts, but I have no way of verifying nor seeing the outputs unless I commit to run a very computationally expensive model. As I mentioned, this notebook could serve to highlight and showcase the modelling pipeline in a coherent way, by just importing the files and having everything tidy, hiding all unnecessary complexity, and displaying only the top-level functions from the API.

Answer: Thank you for these suggestions! We adjusted the repository according to some of them. It now includes a Jupyter notebook that demonstrates the workflow of applying the model to a region of interest (Hardangerjøkulen, 2022) with explanations and visualisation of the intermediate and final results.

Regarding the folder structure, we moved some of the files (related to data and models) to internal folders. Yet, we kept all the entry points (.py and .ipynb files that users are expected to run) in the root directory as it is a well-established practice in organising software packages like this.

The same applies to the existing folders---they are not just folders but also Python modules. Most of them (except maybe configs) are easily reusable in other projects, thus, there is an optimal structure based on hierarchy between them and the software engineering principles, which we are trying to follow. Because of that, we did not rearrange the modules.

Please see the changes at https://github.com/konstantin-a-maslov/scalable_glacier_mapping . The demo itself is at https://github.com/konstantin-a-maslov/scalable_glacier_mapping/blob/main/demo.ipynb , also a link was added to README.

Response to Reviewer #3

The authors have improved their manuscript to be more accessible to a broad readership, moving several methodological elements containing machine-learning jargon to the Methods section, and renaming some “own-coined” terminology for clarity based on comments also echoed by other referees. The quality of both text and figures is still high, and now the manuscript is also more generally accessible.

Additionally, the authors have partly addressed my main comment about investigating quantitatively if their automated prediction is on par with human expert delineation, which I think has greatly helped re-focused the study’s conclusions into a clear applicative result in terms of mapping accuracy.

Answer: We are thankful for your comments and positive feedback, which greatly contributed to the improvement of the manuscript!

However, the authors did not address the core of that comment, that is that GLIMS outlines cannot truly serve as a reference for estimating accuracy, and overlooked some of my propositions to remedy that (that seem to have been misunderstood). I think that this limitation is still a strong one, but could be fairly simply remedied by gathering a small dataset of very-high resolution outlines (and only outlines) that exist in different regions with local aerial surveys, which are small datasets that authors are often willing to share. Those would be largely error-free compared to GLIMS, and would make the perfect dataset (even if a small sample) for robust “validation”, to fully back up the statement now made by the authors that their predictions are “on par” with human expert delineation.

Answer: Thanks for clarifying this point, it was indeed partially misunderstood in the first revision. Therefore, we now included the comparison with the Swiss Glacier Inventory 2016 (SGI2016; Linsbauer et al., 2021) derived from 0.5 m resolution data, as it was suggested as golden standard by the reviewer before. Yet, we remain sceptical about whether one can call any inventories “error-free”. For a more detailed discussion, please see the answer to your specific comment below.

1/ Short comment on claims on impact in rebuttal

The authors repeatedly insist in their rebuttal to not forget that the advances from their methods are mostly that mapped outlines are consistent and scalable without the use of many human resources (as they are automated), which is in itself a great value. While of this is of course true, based on the state of the field, a new glacier outlines dataset would likely have limited scientific impact if its accuracy is not rigorously proven to match existing human-based methods. And this, even if it provides new multi-temporal outlines with less effort.

The reasons being, for instance, that:

- Current glacier mass change estimates are not hampered that much by errors in modern glacier outlines due to temporal differences, as elevation change on bare-rock near the margin of the glacier (or where the glacier has retreated during a period) is nearly zero, so errors due to delineation are actually minimal (and knowingly over-estimated in the mass change community by applying a large % of error); Errors in delineation do affect specific mass

change rates a lot (= mass change divided per the glacier area) but those are usually only used to interpret the climatic signal, and not for providing accurate estimation of direct volume/mass changes used for sea-level rise, river runoff, etc,

- Same reasoning for velocity estimates, where measured velocities near glaciers (or where glaciers have retreated) are close to zero, and so delineation errors also have a minimal impact on velocity errors, even if they are not from the right time period.

In short, the accuracy of the derived outlines remains the most important criteria to hope supersede current inventories with GlaViTa, and remains definitely more important than the perspective to produce multi-temporal outlines thanks to the scalability of the method.

In regards to the above, I think several statements are currently a bit over-stated by the authors in the introduced paragraphs at lines 58-80 and need to be revised.

Answer: We fully agree with the statement that new multitemporal inventories are only useful and picked up by the community when they have sufficient quality. We, therefore, softened the tone in the introduction to avoid overselling the importance of the work and to emphasize that the quality of the outlines is a crucial aspect of newly generated inventories. (Lines 55, 62-63, 75-76). At the same time, we think that the reviewer slightly underestimates the errors introduced by glacier outlines. For instance, according to our experience, the datasets based on RGI can overestimate glacier volumes by up to 30% and more because of the outline error, as seen in some glaciers in, e.g., Adventdalen, Svalbard. This observation comes from comparing the Millan et al., 2022 dataset (RGI-based, with modelling focused on the period 2017–2018) trimmed to match the 2020 inventory (NPI) and assessing the excess ice volume resulting from outdated or inaccurate outlines. In fact, larger errors could be anticipated since this simple trimming approach does not account for boundary conditions where ice thickness at the 'updated' boundaries is significantly above zero. These errors may be less critical for larger ice complexes and global studies but become more and more important for studying fast-changing regions such as the Arctic and more localised effects.

2/ Main comment: adding an error-free validation dataset from “ground truth” data

In my opinion, the biggest limitation of this study remains the same, which is that the test dataset is a subset of GLIMS outlines that themselves have large errors, and so, despite the large sample tested, it is hard to truly assess the accuracy of the algorithm and interpret its accuracy.

Answer: GLIMS data, and especially its subset RGI has been used in a number of highly cited studies (Hugonnet et al, 2021; Millan et al., 2022; Rounce et al., 2023, ...) and is the best available database to work with. Yet, we agree with the reviewer that glacier outlines from the GLIMS database may contain substantial errors. Therefore the GLIMS data used in this study underwent a second quality control by the authors, in which we didn't find any severe error (debris-covered tongues, however, may remain a potential source of inaccuracies within our dataset). After all, we are confident that, despite the known and cited limitations of satellite-derived inventories, we worked with the best available dataset for our purpose and that we are in fact able to access and interpret its accuracy, both in numbers and visually. Furthermore, we show that the method is accurate enough to resolve decadal glacier area

changes and hence will enable us towards fulfilling the GSOC standards for estimating regional glacier area change for all glacierised regions in the near future.

We fully agree that it is nevertheless useful to perform an additional experiment with a very-high-resolution inventory to underpin our claims that we are on par with expert delineation even when very high resolution is used as a reference.

As the authors state themselves in their rebuttal: “However, as we only compare our results with manually derived inventories, we cannot claim any improvement by the model over them in terms of accuracy.”

In response to my previous review, the authors introduced two new evaluation exercises:

1. Comparing GlaViTa to simple band-ratio methods,
2. Evaluating GlaViTa errors with new metrics in a subset of the test dataset. They use instead the statistical mean and spread of distance deviation instead of the IoU, which I also believe are much better suited to evaluating the structure of errors in this type of predictions and are a great addition.

Those are definitely valuable, but do not address the above issue.

My recommendation (that seems to have been misunderstood the first time) to address this issue remains the same:

The authors could gather independent, high-resolution outline datasets (often mapped from 1-m or even 0.5-m LiDAR imagery) that do exist in many countries (Switzerland, Scandinavia, Canada, etc), often linked to national surveying efforts. This would provide a largely error-free reference to test GlaViTa independently of GLIMS. The text cited by the authors in their rebuttal from Paul et al. (2013) on errors being independent of resolution relates to satellite imagery and is really not applicable to aerial-based lidar imagery, which is not only much more resolved (sub-meter) but also much more precise (almost no short-scale error correlations, contrary to satellite imagery).

To clarify the potential misunderstanding further: GlaViTa would NOT need to use any of the LiDAR imagery related to these outlines to predict them, but simply use the temporally-closest satellite imagery it is already using and use this outline instead of the GLIMS outline as the ground-truth reference at that time period. Consequently, only outlines would be needed, and many have been digitized throughout the community with authors often happy to share the data (especially if it is just a shapefile and not the lidar imagery, it really is a tiny time investment). A small dataset sample of a dozen glaciers would already be a great independent validation.

This type of validation with very high resolution data such as LiDAR is exactly what is done in other types of remote sensing estimations (e.g., glacier mass change studies), and would be as beneficial here as it is in these other studies to validate the accuracy of the delineation estimates, including the average IoU (or better, the distance deviation metrics), as well as the range of predicted uncertainties.

Answer: We appreciate the reviewer’s suggestion and addressed this concern by incorporating an additional validation using SGI2016 (Linsbauer et al., 2021), which is

derived from sub-meter resolution aerial imagery and spans from 2013—2018. We specifically selected an area within the Swiss Alps (covered by one Sentinel-2 tile) and collected corresponding satellite data from the summer of 2016 (we only expect minor changes within the potential temporal baseline), including Sentinel-2 and Sentinel-1 imagery, as well as Cop30DEM tiles. These data were processed through the GlaViTU model trained globally on Optical+DEM+InSAR data.

The results showed a strong correspondence between the predicted outlines and the reference data (IoU = 0.865), with 74.74% of misclassified pixels falling within the 95%-confidence bands and an expected calibration error of 0.00231. The total area was underestimated by -6.39% (varying from -29.34% to 12.84% if taking into account the confidence bands). We further conducted a detailed analysis using a subsample of 15 glaciers, comparing area and distance deviations, which were largely consistent with previous results, except for one glacier (Wallenburfirn) where debris-covered tongue contributed to a higher area error of -30.44%. Part of the tongue was within the confidence bands, and the relative area deviation would be -1.51% if the bands were included. Yet, this discrepancy is still in the range of human expert uncertainty when interpreting satellite images (Paul et al., 2013). For 10 out of 15 glaciers, the mean distance is smaller than 3 pixels and the median distance is smaller than 1 pixel. Notably, area deviation is mostly negative, which we mainly attribute to the treatment of lateral moraines inconsistent with GLIMS (SGI2016 tends to include more of them).

Glacier delineation accuracy for a subsample of 15 glaciers from SGI2016. The percentage of debris cover is reported as the fraction of glacier area not classified by the band ratio method within the reference outlines. Ice divides were copied from the reference data.

Glacier	Debris coverage, %	Pixel size, m	Area, km ²		Area deviation, km ² %		Distance deviation, m			IoU
			Reference	Predicted	Mean	Median	95 th percentile			
Lötschegletscher	39.70	10	0.731	0.691	-0.040	-5.54	20.02	12.49	70.46	0.852
Gamchigletscher	65.16	10	1.065	0.953	-0.112	-10.54	45.64	25.77	158.52	0.626
Wallenburfirn	33.53	10	1.387	0.965	-0.422	-30.44	62.04	14.68	289.95	0.661
Chelengletscher	8.04	10	1.764	1.701	-0.063	-3.54	12.41	5.15	58.45	0.929
Weissmiesgletscher	45.45	10	1.920	1.705	-0.215	-11.19	39.06	12.49	178.50	0.813
Alpjergletscher	3.80	10	2.091	2.112	+0.022	+1.04	15.47	6.63	60.54	0.917
Breithorngletscher	32.14	10	2.529	2.569	+0.040	+1.59	14.00	7.89	49.91	0.922
Oberaletschgletscher	27.59	10	3.949	3.591	-0.358	-9.06	33.05	11.63	137.87	0.826
Langgletscher	14.83	10	8.000	7.643	-0.357	-4.46	31.34	8.61	135.92	0.902
Kanderfirn N	8.50	10	11.961	12.027	+0.066	+0.55	20.65	8.52	79.62	0.960
Hüfirn	5.94	10	12.622	12.112	-0.510	-4.04	19.47	8.12	80.83	0.948
Triftgletscher	4.30	10	14.548	14.378	-0.171	-1.18	18.31	6.37	83.79	0.954
Rhonegletscher	5.60	10	14.626	14.038	-0.588	-4.02	25.58	8.22	125.52	0.949
Unteraargletscher	41.66	10	22.681	21.311	-1.371	-6.04	28.26	10.01	123.66	0.897
Aletschgletscher	11.31	10	78.436	77.014	-1.421	-1.81	20.82	7.62	94.11	0.952

Weissmiesgletscher

Interestingly, our results often were somewhere ‘in between’ the closest GLIMS records (2015 inventory; Paul et al., 2020) and SGI2016 (as seen e.g. in the figure on the right), even given that the GLIMS database was used for training.

As an important note, we acknowledge that SGI2016, despite being derived from high-resolution imagery, is also not without its limitations. Linsbauer et al., 2021 report the results of their own round-robin experiments for SGI2016 including 5 experts: “... for bare-ice glaciers, where the glacier ice directly meets bare bedrock, digitized glacier margins lie in general within 2–10 m of horizontal distance. Only in some exceptional cases they differed by up to 100 m because of in-/excluding small branches (Figure 12A [we copied this figure, see below], cf. southern spike) or due to snow cover or clouds (Figure 12C). For debris-covered glacier ice on sedimentary beds the deviations of the outlines are often somewhat larger (5–50 m), i.e. the different interpretations of the glacier boundaries do not agree everywhere, but in most cases match well (Figure 12D). The largest variability between digitized outlines has been found for a very small glacier in a shadowed, snow-covered north face (Figure 12B) with a standard deviation of inferred glacier area of 23.8%. For all other of the 15 analyzed glaciers covering different size classes and characteristics, the standard deviation lies between 0.3 and 7.1%.” Note also that Linsbauer et al., 2021 report one standard deviation of area while we report relative area differences.

Round-robin experiment results by five experts from Linsbauer et al., 2021.

These numbers are also in line with the findings from Paul et al., 2013: “The boundary of the ice can, despite the high spatial resolution, only be roughly estimated” and significant relative area differences were found when digitized by several experts: “For the glaciers digitized on aerial photography, the STD is 1.6–8.8% (mean 3.6%).” They conclude: “However, when using such high resolution data as a base for a digitization, the precision of the derived glacier area is not necessarily higher (e.g. due to low contrast or difficult interpretation).”

Nevertheless, we believe that this additional validation provides a meaningful and independent assesement (sic) of the GlaViTU model, offering further insights into its effectiveness and robustness. These experiments were summarised in S.2.

3/ Minor comment: Adjusting the title

I don't think the title currently does any favour to the study: “Towards” somewhat implies a next study or a step forward which makes the nature of the findings in this study immediately

unclear. A first reader wants to know directly what is the main advance of this study specifically. Additionally, this is a case study (even though spread out through different regions), not a global study. For clarity, the “global” could be kept for a later study.

For instance, the following title seems more appropriate:

“Scalable deep learning glacier mapping using open Earth Observation data matches the accuracy of manual delineation”

Answer: Agreed, thank you for this suggestion. We adjusted the title, now it reads “Scalable glacier mapping using deep learning and open Earth observation data matches the accuracy of manual delineation.”

References:

- Linsbauer, A. et al. *The New Swiss Glacier Inventory SGI2016: From a Topographical to a Glaciological Dataset. Frontiers in Earth Science* 9, 704189. issn:22966463 (Oct. 2021).
- Paul, F. et al. *On the accuracy of glacier outlines derived from remote-sensing data. Annals of Glaciology* 54, 171–182. issn: 02603055 (July 2013).
- Paul, F. et al. *Glacier shrinkage in the Alps continues unabated as revealed by a new glacier inventory from Sentinel-2. Earth System Science Data* 12, 1805–1821.1241 issn: 18663516 (3 Aug. 2020)
- Hugonnet, R. et al. *Accelerated global glacier mass loss in the early twenty-first century. Nature* 592, 726–731. issn: 14764687 (7856 Apr. 2021).
- Millan, R., Mouginit, J., Rabatel, A. & Morlighem, M. *Ice velocity and thickness of the world’s glaciers. Nature Geoscience* 15, 124–129. issn: 1752-0908. <https://www.nature.com/articles/s41561-021-00885-z> (2 Feb. 2022).
- Rounce, D. R. et al. *Global glacier change in the 21st century: Every increase in temperature matters. Science* 379, 78–83. issn: 10959203. <https://www.science.org/doi/10.1126/science.abo1324> (6627 Jan. 2023).

Reviewer #3 (Remarks on code availability):

The repository has improved since the last round of review, with what seems like sufficient details in the README to help other users use the software. I have not run the software.

Answer: Thank you! Meanwhile, we further improved the repository. Now it contains a Jupyter notebook demonstrating the workflow of applying a pretrained model to a region of interest. Please see at https://github.com/konstantin-a-maslov/scalable_glacier_mapping

Response Letter—Globally Scalable Glacier Mapping by Deep Learning Matches Expert Delineation Accuracy

Response to Reviewer #1:

The authors have addressed all my remaining comments, and I feel the manuscript is ready for publication.

Answer: We thank you for your thoughtful comments and are grateful for your positive assessment that the manuscript is now ready for publication! Your feedback has been invaluable in refining our manuscript.

My only final comments are regarding the GitHub repository, which I have written separately in the code section of the review.

While the repository has greatly improved, I still feel it could benefit from a bit more structuring in order to make it more user-friendly and increase the chances of being reused by other researchers. I would suggest the folders and files to be better structured. I understand that the authors have chosen to show folders based on modules. I still think it's best to have a unique folder per package, which inside groups all the folders of the modules. I will not insist on this since it can be quite personal. Nonetheless, I think it would still be a good idea to group all notebooks showcasing the code in a single folder at root level. This can be called `notebooks` for example. Finally, I would also make sure to directly reference the demo Jupyter notebook quite high in the README, so new users can find it easily and get a good overview of how the code works.

Answer: We appreciate these suggestions. We duplicated the link to the demonstration notebook higher in the readme file to make it more visible according to a reviewer's point, please see the changes at https://github.com/konstantin-a-maslov/scalable_glacier_mapping/tree/v1.0. As for the rest, following our previous argumentation, we kept the folder structure the same. This structure, however, has proven to be quite user-friendly as was evident during the 3rd Machine Learning in Glaciology workshop, where participants managed to reuse our codebase with minimal guidance.

Response to Reviewer #3

The authors have addressed my previous comments in their entirety and satisfactorily, in particular by performing a new comparison to independent, high-resolution glacier outlines to further back up their study's main claim of being "on par" with manual delineation.

This new comparison is rigorous and its findings are in-line with the authors' previous conclusions, reaching even better accuracy reached than on the other "independent dataset", comparable to that of manual delineation. Additionally, the authors have added the right level of material to both the main text and supplementary to describe this additional analysis, and adjusted the introduction and title to better reflect the study's impact and content.

Thus, in my opinion, this exciting study is ready for publication!

Answer: Thank you for your positive feedback on our revised manuscript and the new analysis we added. We are glad to hear that you find the work ready for publication. We appreciate your help throughout this process.

The repository is still of good quality, and even further improved.

Answer: Thank you once again for your thorough review.

Review of "Towards Global Glacier Mapping with Deep Learning and Open Earth Observation Data" by Maslov et al.

1 General comments

Maslov et al. present a new approach to automatically map glacier extents from multi-source remote sensing data, based on a convolutional-transformer deep learning model. The authors tackle an important problem present in the glaciological community, particularly aiming at improving the temporal resolution and homogeneity of glacier inventories, which would be an important milestone to improve the validation of regional-to-global glacier models. They do so by designing a custom architecture, and by thoroughly assessing different training strategies. This technical complexity represents an important improvement with respect to previous more simple and naive efforts, based on readily-available architectures. The scale of the problem also represents a meaningful improvement compared to the current literature, with previous studies only focusing on reduced regions. For this, this study has a great potential, particularly as an open-source tool which can be progressively refined as a community effort, as suggested by the authors. The fact that the authors share their code in a reproducible manner in GitHub, with a properly structured repository, is a great plus. Moreover, once they share the training dataset, if done correctly, this can serve as a good benchmark for community efforts to try to build on top of this work. This represents the smartest and fastest way to tackle this problem head on, following the principles of open science.

Nonetheless, I believe this study is missing further depth into the analyses of some of the choices and consequences of the results obtained, as well as the bigger picture and the potential of moving towards a reliable and automated method for extracting glacier outlines globally. The paper sometimes feels rather factual and superficial, and it lacks scientific depth beyond the purely technical aspects of the development and training of the model. I have few comments on the technical aspects of the model, which I believe to be generally very solid and well evaluated. However, I would like to see certain aspects of the paper, particularly in the discussion, further developed in order to clearly show both the real scientific potential and impact that this can have for the glaciological and wider Earth observation community, as well as a better understanding on the choices and reasons behind some of the obtained results.

I will address my general comments in GC points, and then I will move to specific line-by-line comments.

1.1 GC1: Lack of perspective and "bigger picture"

Due to the wide audience and nature of this journal, I believe it would be interesting to further develop the potential and impact of this work, first for the glaciological community, and then for the Earth observation community.

Starting from the introduction, I believe the current issue with glacier inventories is clearly explained. Nonetheless, I think it would be interesting to better explain why this matters for scientific efforts specifically. There is a lot to be said about how glacier outlines are the baseline product for almost all regional-to-large scale glaciological studies. First of all, remote sensing researchers deriving many glacier products (e.g. ice surface velocities or geodetic mass balance), rely on these outlines. Any biases in those outlines impact all the products used by the community. But that is just the beginning of the workflow, since then modellers use those datasets to force their models and make assumptions about physical processes or the past or future evolution of glaciers. Moreover, having multi-temporal consistent outlines could further help constrain the evolution and trajectory of glacier models during calibration for past periods. While this is briefly mentioned in the introduction, I think not enough emphasis is put on the importance of this work, somehow diminishing the impact for the wider community.

Some aspects of this could also go in the discussion part, particularly regarding the impact of this work, or at least the potential of tackling this problem in such a general and flexible manner. Another aspect that is missing, is the impact of this work in the Earth observation community. How does this compare to similar efforts for other Earth observation communities outside glaciology? Do any of the lessons learnt here transfer elsewhere? I would further focus on the wider impact of this work and how this could be helpful or impactful to the wider community.

1.2 GC2: Limited scientific interpretation of the results

Another aspect that is lacking in this study is more depth in the interpretation of some of the results. While the technical aspects are solid, and the training strategy comparing many approaches is very thorough, one has the impression that some very interesting results are barely analysed or commented, keeping the focus purely on the technical aspects of the machine learning model training. For instance, the insights on the contribution and importance of thermal and SAR are quite limited, mostly just describing how they impact model performance, without digging too much into the physical reasons. Since the bad performance of thermal data comes quite as a surprise, I would appreciate if the authors could further explain why they think this harms model performance, and following this analysis, how it could be adapted in order to correctly leverage this dataset. Thermal data is widely used to study debris covered glaciers, so this somehow contradictory results should be correctly argued and explained. The same goes with SAR, but in a lesser degree. It would be interesting to have further insight on why SAR backscatter and InSAR coherence do help to map glacier termini and artefacts along coastlines, beyond obvious reasons related to cloud penetration.

1.3 GC3: Naming and explanation of testing strategies

Another aspect that could be improved in the study is the naming and explanation of the testing strategies (i.e. "Multitemporal global-scale generalisation"). Two strategies are presented: (1) a test set based on tiles, and (2) a test set based on different regions and time periods (called "standalone"). First of all, I find the names quite confusing, especially with the "standalone" one. I think the authors should better explain each one of them.

If I understood correctly, strategy #1 is the classic approach of randomly splitting the dataset into train, validation and test. Therefore, the test tiles are picked randomly without respecting any spatiotemporal structures in the data. I would be inclined to call this "Randomized test". Whereas strategy #2 is focusing on smaller regions and isolating specific spatial regions and temporal periods for the test dataset. For this case, I would be inclined to call this "Spatiotemporal test". This better reflects the main differences in both approaches, since both approaches are in fact "standalone", as any out-of-sample test dataset should be. Finally, the large difference in dataset sizes between both approaches makes me wonder how reliable are the results of strategy #2. It would be interesting to provide a metric on the number of glaciers or polygons included in each dataset, in order to better understand the robustness of these results. This is seen a little bit in Fig. S1, where there is a much smaller number of data points for the "standalone" approach.

1.4 GC4: Example in the GitHub repository

While the repository is work in progress and it is already quite complete, it is hard to reproduce the results due to the high computational requirements of the model. Neither forking the repository myself and running it in local, or even attempting to host this on Binder yourselves are not feasible.

For that, I would encourage the authors to provide a Jupyter notebook with an example (e.g. with the 10% dataset), which shows the baseline workflow on how the model is loaded, trained, and how some results/plots are obtained. Since notebooks can be browsed directly on GitHub, this could provide an easy-to-share showcase demo for people wanting to see how the model works. The notebook could be directly linked in the README for easy access.

2 Specific comments

- **Abstract** "...difficult-to-classify debris...": this sentence is a bit confusing. You are not trying to classify debris, but debris-covered ice, or glaciers. I would rephrase this.
- **L122** The IoU acronym has never been introduced before. I would briefly explain what it is.
- **L140** Why is *DeepLabv3+/ResNeSt-101* chosen as a baseline model for comparison? This should at the very least be explained, as it comes quite out of the blue.
- **L152** Indeed, hardcoding spatial coordinates like this might introduces biases. What were those biases in your case?

- **L153-156** See GC3. Here I would also point the reader to the place in the Methods where this is explained in detail. Overall, this should be done throughout the paper to avoid vagueness and help the reader find the meaningful details, especially for a rather technical paper like this one.
- **L159-162** Could you specify how many input features you actually feed into the model? E.g. for optical, do you feed multiple bands? For SAR, I guess it's both intensity and coherence. What about backscatter? Besides specifying the number of input features, I would also refer to a figure/table which gives more details about that.
This made me wonder: did you try combining the four types of sources: optical, DEM, thermal and SAR? Could training with both thermal and SAR at the same time help in some way?
- **L196-207** See GC2. These results are interesting and meaningful from a wider perspective and a physical/scientific point of view. I would expand this and go deeper in the interpretation in the discussion section.
- **Fig. 1** I see 2 losses in the architecture, but I don't recall having read an explanation about the role of each. Where is this explained? This should have its place in the Methods.
- **Fig. 2** By looking at these tiles I cannot help see that many of them give a very partial view of glaciers, with only a small fraction of them. Have you tried different sizes of tiles for the training? Did you try different strategies to try to maximize glacier coverage within each tile? Have you seen if having a complete view of the glacier in a tile helps with the mapping? Since you are using attention, this could very well be the case. Could you please comment on these aspects?
- **L244** In the text you mention "tiles" directly. For readers not familiar with computer vision and CNNs, I would introduce this concept.
- **L248** See GC3. At this point it was quite confusing to follow along the explanations, since it is not very clear what each strategy is and how they differ. I would make sure that this is made clear before carrying on with explaining the results.
- **Fig. 3** For this sort of figures, I would explain more in detail what the figure tries to show in the legend. The same goes for Fig. 1.
- **L301-304** See GC2. Here you correctly point out the outcome of the results, but again, I would go deeper into the analysis in the discussion section.
- **L310-322** Same as the previous one.
- **L329** "Monte-Carlo dropout" and "plain softmax" need references, and at least some brief introduction. I would also point the reader to the place in the Methods where this is explained in detail.

- **L388** Can be or could be? Are these perspectives or actual possibilities in the current framework that is being presented. This should be made clear.
- **L391** Same as the previous one.
- **L451-458** See GC2. These are really interesting results. Please go deeper and interpret them in the discussion.
- **L529-536** I fully endorse this community-based perspective built on open-source software. You provide the baseline model to build upon with the help of the community.
- **Fig. 4** This figure has a lot of potential but in my opinion it is currently underutilized. I would add the percentage of glacier coverage for each region, both in terms of number of glaciers and surface area, and I would also include the total (global) number of glacier coverage (9% according to you), somewhere in the figure.

Moreover, it would be interesting to add some sort of pie chart indicating the distribution of clean ice, debris-covered and marine-terminating glaciers. And finally, including the performance of your model(s) - perhaps only the best one - for each region could also provide an overview of how it changes with respect to the type of glaciers.

Again, I would make proper use of the legend to explain the figure. It should be standalone, so a reader who skims through the paper can have an idea of what is going on.

- **L580-582** Did you balance the appearance of clean ice, debris-covered and ice mélange in each one of these datasets?
- **L608-610** When performing tests for unseen years, were the input features (e.g. DEM, optical) from that new date as well? or only part of them (e.g. optical)?
- **L662-664** Later on you explain how you use these cycles (e.g. to fine-tune global models). Please explain the general purpose of these cycles here, without going into the details you explain later on in the Strategies section.
- **L665-667** Do you mean after each one of these cycles? It is unclear why do you use them.
- **L675** While it is good to have all these details here, I found it was hard to understand the nuances of each approach in the main text. I'd rephrase how this is explained there in order to make it clear for the reader, and I would also clearly point the reader to where each thing is explained more in detail in the Methods section.
- **L713-714** Please provide a reference.
- **L745-746** I guess that the parameters of this FFNN were optimized in conjunction with the other parameters of the architecture?
- **L760** I would make a direct reference to this section in the main text to guide the reader.

- **L767** I would provide a little bit more context on what this is.
- **Fig. S1** Why are there so few "standalone"/spatiotemporal data points? It would be interesting to provide a number of independent test and standalone data points, to understand the robustness of the metrics. I guess each point is a region? Since regions are probably larger than tiles, I am not sure if this direct comparison is meaningful.
- **Fig. S5** Something I've realised from your predicted results is that sometimes there can be holes or gaps in a glacier tongue (e.g. Fig. S5e). Such artefacts are mostly 100% unrealistic, since due to glacier ice flow dynamics, it is practically impossible to have such gaps, especially in areas with a continuous ice flow. For that, an automatic post-processing to eliminate such artefacts could potentially improve model performance with a simple solution.
- **Fig. S11** This figure serves to show the spatial testing strategy, but there is no information about time. I would indicate how the cross-testing has been implemented for each one of these tiles, where the model is trained on all but one, and then evaluated in test on a single tile based on an unseen region. Moreover, giving dates or a timeline of how these different scenes at different time snapshots are used for temporal cross-testing would be useful.

Review of "Towards Global Glacier Mapping with Deep Learning and Open Earth Observation Data" by Maslov et al.

The authors have substantially updated the manuscript, following most of my comments as well as other input from the rest of the reviewers. The manuscript has gained in depth, particularly in some scientific analyses that I asked for; while some technicalities and complexity have been removed from the main text in order to better suit the journal format of Nature Communications. Nonetheless, I still feel that the manuscript needs a few modifications in order to improve its readability and to convey the scientific message of the paper in a clearer manner. As I mentioned in my previous review, the technical aspects of the paper are solid, but the message needs to be properly adapted and the story-line streamlined in order to fit in the narrative of this journal.

I have listed different comments across the paper, which relate to the above-mentioned topics. Overall, my comments go in the direction of these last changes, in order to improve the scientific message and to remove unnecessary technical details from the main text.

Once these changes are addressed, I don't foresee any more changes blocking publication from my side.

- L205: It should read "We propose ...".
- L369-398: While the addition of this analysis for small glaciers is interesting, it suffers again from too many technical details which dilute the flow and tone of the paper. I would suggest keeping the most important results in there, focusing on the **scientific** implications, and move all the technical details to the methods or the supplementary material.
- Fig. 1. I see you have added a little bit more explanations in the figure, but this figure remains highly technical for such a wide audience. Only computer vision experts will make sense of this figure, so I suggest two things: (1) Update the figure by adding envelopes or some visual trick to distinguish the role of each part of the network. The goal here would be to "translate" each part of the architecture to the scientific/applied task which can be understandable by a wider audience; (2) Add these extra explanations in the caption as well. This figure should be much more self-explanatory if it has to remain in the main text.
- Regarding my previous comment on the partial coverage of tiles for glaciers. I understand that you are hitting the memory limit of GPUs with the largest possible tiles.

Nonetheless, could you verify if the performance is better for already existing tiles that happen to cover a full glacier? This could be an interesting hint. You could relate this to the metrics you have shown where your model performance is much better for large glaciers. This is interesting, because for large glaciers you will almost never have a complete view of the glacier in the tile, so from a technical point of view, it could become equivalent to just seeing a smaller glacier. It would be interesting to disentangle this and to understand exactly why you are obtaining these results.

- Fig. 3. I see you have expanded the legend, but this figure still feels way too obscure. Again, I think you should really try to make Figures as self-explanatory as possible. Try to design them in a way so readers quickly browsing the paper can make sense of that topic even without reading the full text. You mention before and after calibration. Calibration of what? Model parameters? Hyperparameters? Also, explain the results. Don't let readers interpret them by themselves only. What are you trying to convey here? What relationships can we see, and why is this important?
- GitHub repository: While the repository is quite complete, and the README covers quite a lot details, I still feel that the files are quite spread out and it generally lacks a bit of structure. Instead of dumping so many files at the root level of the repository, I would encourage to further structure this into folders, and to clearly separate the internals from the scripts and API that the user needs to use. This means, for example, to place the scripts for the workflow in a single folder, and to reduce the already existing number of folders by further grouping them into folders. While I understand that for pure processing reasons it is much better to have the files as .py scripts, I still think it would be nice to have a Jupyter notebook already executed, just for showcasing purposes. I have browsed and opened the scripts, but I have no way of verifying nor seeing the outputs unless I commit to run a very computationally expensive model. As I mentioned, this notebook could serve to highlight and showcase the modelling pipeline in a coherent way, by just importing the files and having everything tidy, hiding all unnecessary complexity, and displaying only the top-level functions from the API.